# Lossy Common Information in a Learnable Gray-Wyner Network

**Anderson de Andrade, Alon Harell & Ivan V. Bajić**
School of Engineering Science
Simon Fraser University
Burnaby, BC, Canada
`{anderson_de_andrade,alon_harell,ibajic}@sfu.ca`

## Abstract

Many computer vision tasks share substantial overlapping information, yet conventional codecs tend to ignore this, leading to redundant and inefficient representations. The Gray-Wyner network, a classical concept from information theory, offers a principled framework for separating common and task-specific information. Inspired by this idea, we develop a learnable three-channel codec that disentangles shared information from task-specific details across multiple vision tasks. We characterize the limits of this approach through the notion of lossy common information, and propose an optimization objective that balances inherent tradeoffs in learning such representations. Through comparisons of three codec architectures on two-task scenarios spanning six vision benchmarks, we demonstrate that our approach substantially reduces redundancy and consistently outperforms independent coding. These results highlight the practical value of revisiting Gray-Wyner theory in modern machine learning contexts, bridging classic information theory with task-driven representation learning.

## 1 Introduction

It is often the case that a machine task – classification, recognition, etc. – requires only a subset of the information provided as input. We can interpret neural networks as processes that discard irrelevant information from a signal so that its predictions are in a probability space similar to that of the target (Tishby & Zaslavsky, 2015). In multi-task settings, the same input is used to perform different tasks. These tasks have semantically different targets and hence might require different subsets of the input information.

Whenever tasks are not performed jointly, isolating the information required for each one is critical to ensure that communication is efficient. For example, when only object detection is needed, it would be efficient for a camera to only transmit the information required so the receiving device can perform that task. If it is then decided that semantic segmentation is needed for the same input, it would be efficient for the camera to only transmit the *additional* information necessary, considering that some relevant information has already been transmitted. On the receiver device, it would be efficient if the information from the first transmission, that is also relevant for semantic segmentation, is isolated, so it can be readily used without overhead.

The line of work in *coding for humans and machines* (Choi & Bajic, 2022) focuses on this problem. It considers an image reconstruction task together with a computer vision task. It is commonly assumed that all the information used by the computer vision task is relevant to the reconstruction task. Thus, only two separate channels (representations) are often designed: a common channel used by both tasks, and a private channel only used by the reconstruction task. In this work, we focus on a pair of tasks that have some common information (CI) between them, but also private (dedicated) information for each task, establishing three channels: a common channel, and two private channels.

We establish in this work that a complete isolation of the common information needed between two tasks is often unattainable. This also occurs in *lossy coding* (compression), where we send less information (*rate*, e.g. *bitrate*), which inevitably performs the tasks with a higher error (*distortion*).

A system must then decide between transmitting some of the non-common information on the common channel, or some of the common information on the private channels. In the former case, the *transmit rate* – the amount of information transmitted when performing both tasks on one device – can be optimal, but the *receive rate* – the amount of information transmitted by performing each task on a different device – will not be optimal. The converse is true in the latter case.

We propose a neural network architecture that is able to separate the common information between two tasks. We compare the proposed architecture against other more intuitive architectures we designed, and include a theoretical justification for their difference in performance. We also propose a loss function that can optimize a codec for the transmit rate, the receive rate, or a tradeoff between both. We show how our methods perform on different pairs of computer vision tasks.

## 2  PREVIOUS WORK

In information theory, *source coding* methods convert a sequence of symbols from an information source into a sequence of bits, allowing data compression. Our work is closely related to the Gray-Wyner Network (GWN) (Gray & Wyner, 1974). It is a source coding problem connecting two sources with two receivers via a common channel and two private channels. The work defines the *Gray-Wyner region* as a set of achievable rates (e.g. bitrates) for the three channels, in both lossy and lossless input reconstruction tasks.

There are multiple notions of common information in literature (Viswanatha et al., 2014), including *mutual information*. In this work, we discuss Wyner's common information (Wyner, 1975) and Gács-Körner (GK) common information (Gacs & Körner, 1973). These quantities are located in the Gray-Wyner achievable region. They are related by a tradeoff between the total transmit and receive rate (Viswanatha et al., 2014).

The focus of seminal work on *learnable image coding* (Theis et al., 2017; Ballé et al., 2018; He et al., 2022) has been image lossy coding (compression) as a single task. They consist of a lossy *analysis transform* (encoder) and a *synthesis transform* (decoder) acting as an autoencoder (Ballé et al., 2018) such that the latent representation is better suited for coding, achieving a lower rate. An *entropy model* uses various types of context to predict the distribution of a target representation. When optimizing for rate-distortion performance (Theis et al., 2017), the entropy model effectively induces an information bottleneck (Tishby et al., 1999) on the target representation. A higher penalty on the probability estimates of the entropy model for the target representation produces lower rates at the expense of a higher task distortion (error).

The work in coding for humans and machines uses similar architectures to those used in learnable image coding. In Choi & Bajic (2022), the output representation of an analysis transform is split in two. The computer vision task uses one representation over a common channel. The image reconstruction task uses the other representation over a private channel in addition to the representation from the common channel. Better rate-distortion performance on the computer vision task was achieved when using two separate analysis transforms, one for each of these channels (Foroutan et al., 2023). However, the common channel had information that was not fully utilized by the reconstruction task. An ad-hoc reconstruction task for the common channel was shown to increase the usability (compatiblity) of the corresponding representations (de Andrade & Bajic, 2024).

Multitask learnable codecs exist in the literature (Chamain et al., 2021; Feng et al., 2022; Guo et al., 2024). They propose one or more common channels to perform several tasks, without private channels. Their rate is optimal only when all the tasks involved are performed jointly.

In representation learning literature, several information-theoretic approaches propose variational autoencoders (VAEs) that learn disentangled representations of a source (Chen et al., 2016; Higgins et al., 2017; Chen et al., 2018b). These approaches are unsupervised in nature and thus do not distill information for a specific task or isolate the common information between tasks. A more related approach (Dubois et al., 2021) proposes a channel that can achieve high performance in a set of predictive tasks, as long as they are invariant under a set of transformations. Although their proposed methods are also unsupervised, the right set of transformations can be used to isolate common information between tasks. However, finding these transformations for non-trivial tasks is an open question. Several variational methods have been proposed to measure the mutual information between two sources (Poole et al., 2019). Although they might seem useful as training objectives,

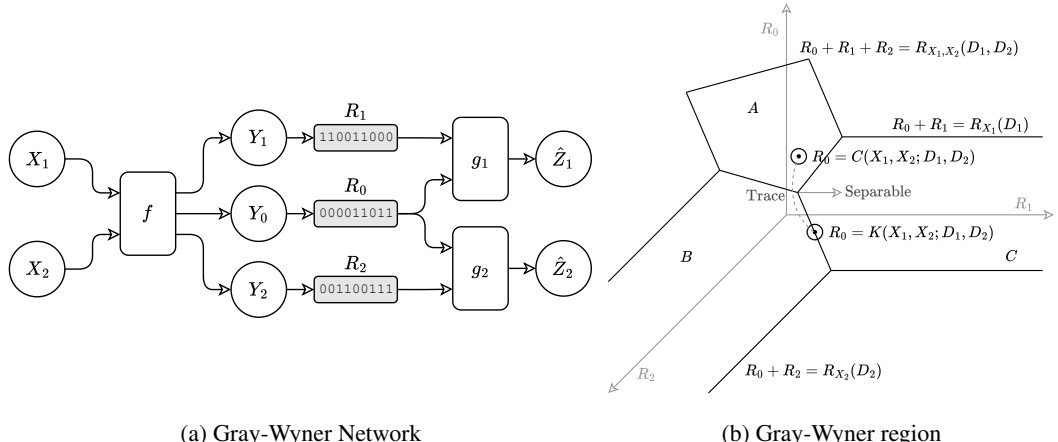

(a) Gray-Wyner Network          (b) Gray-Wyner region

Figure 1: Diagram of the Gray-Wyner Network and a lower bound on its achievable region. The lower bound is given by planes $A$, $B$, and $C$. Plane $A$ corresponds to the *Pangloss*, the contour of the achievable region that achieves $R_{X_1,X_2}(D_1, D_2)$. Points with $R_0 = C(X_1, X_2; D_1, D_2)$ are found on it. Planes $B$ and $C$ achieve $R_0 + R_1 = R_{X_1}(D_1)$ and $R_0 + R_2 = R_{X_2}(D_2)$, respectively. On the intersection of both of these planes, we can find a point with $R_0 = K(X_1, X_2; D_1, D_2)$. A trace connecting both points offers tradeoffs between the transmit and receive rates. If the mutual information between tasks, at distortions $(D_1, D_2)$, is separable, both points coincide at the now achievable *Separable* point. We show that points close to it, whether achievable or not, serve as bounds for both lossy common information measures.

these methods have significant tradeoffs between bias and variance. Moreover, they do not naturally offer the means to code (compress) the resulting representations. Compared against these existing techniques, our work directly addresses the isolation of common information between tasks and its efficient transmission.

## 2.1 PRELIMINARIES

We present the Gray-Wyner Network (Gray & Wyner, 1974) and define the two notions of common information that define the boundaries of the transmit-receive tradeoff on which we want to operate. Let $X_1$ and $X_2$ be two source random variables. In the Gray-Wyner Network, an analysis transform (encoder) $(Y_0, Y_1, Y_2) = f(X_1, X_2)$ produces one common and two private *discrete* representations with corresponding rates $(R_0, R_1, R_2)$. Two synthesis transforms (decoders) $\hat{Z}_1 = g_1(Y_0, Y_1)$ and $\hat{Z}_2 = g_2(Y_0, Y_2)$ predict the targets $Z_1$ and $Z_2$ of two dependent tasks, producing distortions $D_1 = d_1(\hat{Z}_1, Z_1) = \mathbb{E}[\hat{d}_1(\hat{Z}_1, Z_1)]$ and $D_2 = d_2(\hat{Z}_2, Z_2) = \mathbb{E}[\hat{d}_2(\hat{Z}_2, Z_2)]$, respectively. In our context, the distortion functions $\hat{d}_1$ and $\hat{d}_2$ are task losses for targets $Z_1$ and $Z_2$. We assume that one source does not have exclusive information that could assist its non-corresponding task, implying the Markov conditions:

$$Z_2 \leftrightarrow X_2 \leftrightarrow X_1, \qquad\qquad Z_1 \leftrightarrow X_1 \leftrightarrow X_2. \qquad (1)$$

The set of all *achievable* rate tuples $(R_0, R_1, R_2)$ for distortions $(D_1, D_2)$ is given by the Gray-Wyner region $\mathcal{R}_{\mathrm{GW}}(D_1, D_2)$. The contour of this region lower-bounds all other points in this convex set and are considered optimal in that they optimize a particular rate-distortion function.

A rate-distortion function determines the minimal rate that should be communicated over a channel, so that input signals (sources) can be reconstructed without exceeding expected distortions. The joint rate-distortion function (Cover & Thomas, 2006) is given as $R_{X_1,X_2}(D_1, D_2)$. In our context, we define it as the minimum rate required to encode $(\hat{Z}_1, \hat{Z}_2)$ jointly as a function of $(X_1, X_2)$ so that tasks $(\hat{d}_1, \hat{d}_2)$ can be performed with at most $(D_1, D_2)$ distortion. Marginal rate-distortion functions are given as $R_{X_1}(D_1)$ and $R_{X_2}(D_2)$, with similar definitions applying only to one task and one corresponding source. The conditional rate-distortion function $R_{X_1|X_2}(D_1)$ is the minimum rate required to encode a function of $X_1$, such that the task $\hat{d}_1$ can be performed with at most distortion $D_1$, when $X_2$ is available to both the encoder and the decoder.

In this setting, using previous definitions, we define the Wyner's lossy common information as:

$$C(X_1, X_2; D_1, D_2) = \inf I(X_1, X_2; U), \tag{2}$$

where the infimum is over all joint densities $P(X_1, X_2, \hat{Z}_1, \hat{Z}_2, U)$, under these Markov conditions:

$$\hat{Z}_1 \leftrightarrow U \leftrightarrow \hat{Z}_2, \qquad\qquad (X_1, X_2) \leftrightarrow (\hat{Z}_1, \hat{Z}_2) \leftrightarrow U, \tag{3}$$

and where $P(\hat{Z}_1, \hat{Z}_2 | X_1, X_2) \in \mathcal{P}_{D_1, D_2}^{X_1, X_2}$ is any joint distribution that achieves the rate-distortion function at $(D_1, D_2)$, i.e.: $I(X_1, X_2; \hat{Z}_1, \hat{Z}_2) = R_{X_1, X_2}(D_1, D_2)$. The mutual information function $I(\cdot; \cdot)$ measures the dependence between two random variables (Cover & Thomas, 2006). For the conditions in 3 to hold, $U$ must have *at least* the mutual information between $\hat{Z}_1$ and $\hat{Z}_2$.

Gács-Körner lossy common information is defined for our particular setting as:

$$K(X_1, X_2; D_1, D_2) = \sup I(X_1, X_2; V), \tag{4}$$

where the supremum is over all joint densities $P(X_1, X_2, \hat{Z}_1, \hat{Z}_2, V)$, such that the following Markov conditions are met:

$$X_2 \leftrightarrow X_1 \leftrightarrow V, \qquad X_1 \leftrightarrow X_2 \leftrightarrow V, \qquad X_1 \leftrightarrow \hat{Z}_1 \leftrightarrow V, \qquad X_2 \leftrightarrow \hat{Z}_2 \leftrightarrow V, \tag{5}$$

where $P(\hat{Z}_1 | X_1) \in \mathcal{P}_{D_1}^{X_1}$ and $P(\hat{Z}_2 | X_2) \in \mathcal{P}_{D_2}^{X_2}$ are optimal rate-distortion encoders at $D_1$ and $D_2$, respectively, i.e: $I(X_1; \hat{Z}_1) = R_{X_1}(D_1)$ and $I(X_2; \hat{Z}_2) = R_{X_2}(D_2)$. The first two conditions in 5 establish that $V$ cannot have information about $X_1$ or $X_2$ that is not mutual. The last two conditions in 5 establish that all information in $V$ is present *in both* $\hat{Z}_1$ and $\hat{Z}_2$.

Wyner's lossy common information, $C(X_1, X_2; D_1, D_2)$, can be thought of as the least information that must be included in the common channel to avoid the total transmit rate exceeding the optimal $R_{X_1, X_2}(D_1, D_2)$. Conversely, Gács-Körner common information, $K(X_1, X_2; D_1, D_2)$, is the most information that can be included in the common channel while maintaining the optimal receive rate $R_{X_1}(D_1) + R_{X_2}(D_2)$.

The total transmit rate is $R_t = R_0 + R_1 + R_2 \geq R_{X_1, X_2}(D_1, D_2)$ and the total receive rate is $R_r = 2R_0 + R_1 + R_2 \geq R_{X_1}(D_1) + R_{X_2}(D_2)$. Wyner's and Gács-Körner common information can be related through these concepts (Viswanatha et al., 2014). When the transmit rate is optimal, such that $R_t = R_{X_1, X_2}(D_1, D_2)$, we have that the minimum $R_r$ is $R_{X_1, X_2}(D_1, D_2) + C(X_1, X_2; D_1, D_2)$. Such points in the contour of $\mathcal{R}_{GW}(D_1, D_2)$ are achieved when $R_0 = C(X_1, X_2, D_1, D_2)$. When the receive rate is optimal such that $R_r = R_{X_1}(D_1) + R_{X_2}(D_2)$, we have that the minimum $R_t$ is $R_{X_1}(D_1) + R_{X_2}(D_2) - K(X_1, X_2; D_1, D_2)$. This point in the contour of $\mathcal{R}_{GW}(D_1, D_2)$ is achieved when $R_0 = K(X_1, X_2, D_1, D_2)$. This implies that minimizing one type of rate comes at the cost of increasing the other, establishing the transmit-receive tradeoff.

Figure 1 visually describes these preliminary concepts. See Appendix A.1 for more definitions.

## 3 CONTRIBUTIONS

### 3.1 BOUNDS FOR LOSSY COMMON INFORMATION

We extend a result from Wyner (1975) in the lossless setting to the lossy case, showing bounds that separate the two lossy common information terms discussed. The bounds are expressed in terms of interaction information $I(X_1, X_2; \hat{Z}_1^*; \hat{Z}_2^*)$, which is a generalization of the mutual information for more than two variables (Ting, 1962; Yeung, 1991). See Appendix A.1 for its definition.

**Theorem 1.** *Let $\hat{\mathcal{Z}}_{D_1, D_2}^{(t)}$ be the set of tuples $(\hat{Z}_1, \hat{Z}_2)$ that achieve $R_{X_1, X_2}(D_1, D_2)$, and $\hat{\mathcal{Z}}_{D_1, D_2}^{(r)}$ be the set of tuples $(\hat{Z}_1, \hat{Z}_2)$, such that $\hat{Z}_1$ achieves $R_{X_1}(D_1)$, and $\hat{Z}_2$ achieves $R_{X_2}(D_2)$. Then:*

$$K(X_1, X_2; D_1, D_2) \leq \max_{(\hat{Z}_1, \hat{Z}_2) \in \hat{\mathcal{Z}}_{D_1, D_2}^{(r)}} I(X_1, X_2; \hat{Z}_1; \hat{Z}_2) \tag{6}$$

$$\leq \min_{(\hat{Z}_1, \hat{Z}_2) \in \hat{\mathcal{Z}}_{D_1, D_2}^{(t)}} I(X_1, X_2; \hat{Z}_1; \hat{Z}_2) \leq C(X_1, X_2; D_1, D_2). \tag{7}$$

*We have equality everywhere iff the maximum and minimum coincide at $(\hat{Z}_1^*, \hat{Z}_2^*)$, and we can represent $\hat{Z}_1^*$ as $(\hat{Z}_1', W)$ and $\hat{Z}_2^*$ as $(\hat{Z}_2', W)$ such that Conditions 3 and 5 hold for $\hat{Z}_1^*$, $\hat{Z}_2^*$, and $W$.*

*Proof.* See Appendix A.

Theorem 1 allows to interpret the lossy versions of Gács-Körner and Wyner's common information similarly to their lossless counterparts. Any set of tuples $(\hat{Z}_1, \hat{Z}_2)$ that achieve the transmit rate will always have at least the amount of interaction information $I(X_1, X_2; \hat{Z}_1; \hat{Z}_2)$ of any set of tuples that achieve the receive rate. If there is a gap of interaction information between the transmit and receive sets of tuples, exploring the transmit-receive tradeoff will produce tuples outside of those two sets that cover such gap.

The conditions for equality everywhere imply that the two common information terms are the same when the interaction information $I(X_1, X_2; \hat{Z}_1^*; \hat{Z}_2^*)$ is fully separable from private information or *other excess common information* that does not help reach optimal rate-distortion values. Because of the latter reason, this separation can be even more difficult to attain than in the lossless case. To see this, note that Gács-Körner common information in the discrete case is the entropy of the probability distribution of a point in the sample space of $(X_1, X_2)$ belonging to a partition in an ergodic decomposition of the stochastic matrix defining $(X_1, X_2)$ (Gacs & Körner, 1973). This means that to achieve $I(X_1, X_2; \hat{Z}_1^*; \hat{Z}_2^*)$, this information must be separable such that the stochastic matrix $P(\hat{Z}_1, \hat{Z}_2)$ can be written as:

$$\begin{bmatrix} A_1 & & 0 \\ & \ddots & \\ 0 & & A_K \end{bmatrix} \tag{8}$$

where $A_1, ..., A_K$ are probability matrices defined as product of marginals, i.e., with no mutual information. This implies that these random variables can be represented as $\hat{Z}_1^* = (\hat{Z}_1', W)$ and $\hat{Z}_2^* = (\hat{Z}_2', W)$, which, with $W$ being a function of $X_1$ or $X_2$, implies that it can be isolated from $X_1$ or $X_2$. If some of the required mutual information is not separable, it must be left out of $W$ and thus $I(X_1, X_2; \hat{Z}_1^*; \hat{Z}_2^*)$ cannot be produced. The work of Gacs & Körner (1973) shows that Gács-Körner common information is often very small. In fact, it is zero for Gaussian sources with correlation $1 - \rho$ (Viswanatha et al., 2014), which is usually a distribution producing *refinable* results in information theory (Cover & Thomas, 2006). Because we can often expect in practice to have a noticeable gap between the two common information measures discussed, there is a significant motivation to explore the transmit-receive tradeoff.

### 3.2 TRANSMIT-RECEIVE TRADEOFF OPTIMIZATION

As previously discussed, there is a tradeoff between optimizing for the transmit rate, resulting in more information available on the common channel than the mutual information, which increases the receive rate, or optimizing for the receive rate, resulting in less than the mutual information on the common channel, which increases the transmit rate. We propose an objective that can optimize for this tradeoff.

The work of Gray & Wyner (1974) establishes the following objective for optimizing the Gray-Wyner Network:

$$T(\alpha_1, \alpha_2; D_1, D_2) \triangleq \inf \left\{ I(X_1, X_2; Y_0) + \alpha_1 R_{X_1|Y_0}(D_1) + \alpha_2 R_{X_2|Y_0}(D_2) \right\}, \tag{9}$$

where $0 \leq \alpha_1, \alpha_2 \leq 1$, and $\alpha_1 + \alpha_2 \geq 1$, and the infimum is over all probability distributions $P_{Y_0} \in \mathcal{P}_{\mathcal{Y}_0}$ with sample space $\mathcal{Y}_0$. The arguments $\alpha_1$ and $\alpha_2$ specify the transmission cost for each of the three channels. The values of $T(\alpha_1, \alpha_2; D_1, D_2)$ over the domain of $\alpha_1$ and $\alpha_2$ define the contour of the Gray-Wyner region $\mathcal{R}_{GW}(D_1, D_2)$.

Under assumptions suitable to our proposed method, we can express this objective in terms of an optimization over families of functions:

**Theorem 2.** *Assume that $Y_0 = f_0(X_1, X_2); f_0 \in \mathcal{F}_0, Y_1 = f_1(X_1); f_1 \in \mathcal{F}_1, Y_2 = f_2(X_2); f_2 \in \mathcal{F}_2$ are all deterministic functions of their corresponding inputs, and that $g_1 \in \mathcal{G}_1$ and $g_2 \in \mathcal{G}_2$, where $\mathcal{F}_{\{0,1,2\}}$ and $\mathcal{G}_{\{1,2\}}$ are families of functions such that there exits a $f_0, f_1, f_2, g_1,$ and $g_2$ in their respective families that achieve $T(\alpha_1, \alpha_2; D_1, D_2)$. Then:*

$$T(\alpha_1, \alpha_2; D_1, D_2) = \inf \left\{ H(Y_0) + \alpha_1 H(Y_1|Y_0) + \alpha_2 H(Y_2|Y_0) \right\}, \tag{10}$$

*where $H(\cdot)$ is Shannon's entropy function, and $H(\cdot|\cdot)$ is the conditional entropy function (Cover & Thomas, 2006). The infimum is over all $f_0, f_1, f_2, g_1,$ and $g_2$ in their corresponding family of functions, such that we obtain, at most, distortions $D_1$ and $D_2$.*

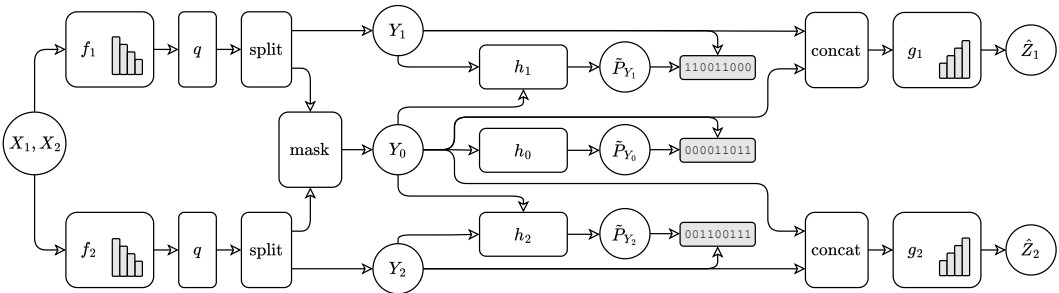

Figure 2: Architecture overview. The parameters for the probability distribution of the target representations are represented as $\tilde{P}_{Y_{\{0,1,2\}}}$. Grey binary blocks denote bitstreams with rates $R_{\{0,1,2\}}$. For the particular choice of analysis and synthesis transforms used in the experimental evaluation, the grey bars on the corresponding blocks indicate the number of downsample or upsample operations.

*Proof.* See Appendix B.

We make use of entropy models to estimate – and also steer – the probability distributions of their target representations. With Theorem 2, we can replace the entropy terms with rate functions $r_{\{0,1,2\}}$ given by entropy models:

$$r_{\{1,2\}}(Y_{\{1,2\}}, Y_0) = -\mathbb{E}_{Y_{\{1,2\}}, Y_0}\left[\log \tilde{P}\left(Y_{\{1,2\}}|Y_0\right)\right], r_0(Y_0) = -\mathbb{E}\left[\log \tilde{P}\left(Y_0\right)\right], \tag{11}$$

where $\tilde{P}$ denotes a probability function assumed for the *quantized* target representations $Y_{\{0,1,2\}}$, implicitly established by the entropy models.

The work of Gray & Wyner (1974) highlights that $T(\alpha_1, \alpha_2; D_1, D_2)$ is difficult to optimize due to the lack of concavity or convexity in $P_{Y_0}$. As such, we propose the Lagrangian relaxation method (Hiriart-Urruty & Lemaréchal, 1993) for the optimization problem in Theorem 2. Relaxing the distortion constraints can help find better solutions in this non-convex space. The Lagrangian relaxation method is used extensively for rate-distortion optimization (Tishby et al., 1999). Due to the convexity of that problem, it is used for convenience, since the method of Lagrange multipliers could also be used. For this problem, however, we have a stronger motivation for its usage.

Assuming that the private channels have the same cost, such that $\alpha_1 = \alpha_2$, replacing the entropy terms in Equation 10 with rate functions, and with the distortion constraints $d_1(\hat{Z}_1, Z_1) \leq D_1$ and $d_2(\hat{Z}_2, Z_2) \leq D_2$, the Lagrangian is given as:

$$\mathcal{L} = \inf\left\{\beta r_0(Y_0) + r_1(Y_1, Y_0) + r_2(Y_2, Y_0) + \lambda_1 d_1(\hat{Z}_1, Z_1) + \lambda_2 d_2(\hat{Z}_2, Z_2)\right\}, \tag{12}$$

where $\beta = 1/\alpha_{\{1,2\}}$, and the infimum is over the same families of functions in Equation 10, with the addition of the families of functions defining the three entropy models. The hyper-parameters $\lambda_1$ and $\lambda_2$ control the rate-distortion tradeoff (Cover & Thomas, 2006).

When $\beta = 1$, we optimize for the transmit rate $R_t$. When $\beta = 2$, we optimize for the receive rate $R_r$. As explained in Sections 2.1 and 3.1, optimizing exclusively for the transmit or the receive rate does not guarantee that the common channel will produce Wyner's or Gács-Körner common information. Therefore, values of $\beta$ outside of the range $(1, 2)$ could result in suboptimal configurations. If we optimize for a combination of both rates, we could attain points on the trace in the contour of the achievable region $\mathcal{R}_{\mathrm{GW}}(D_1, D_2)$, connecting the operational points corresponding to $C(X, D_1, D_2)$ and $K(X, D_1, D_2)$ (Viswanatha et al., 2014). When $\beta = 3/2$, we equally optimize for both the transmit and receive rates. If Theorem 1 holds with equality, an optimal codec optimized for $\beta \in (1, 2)$ achieves both common information measures.

### 3.3 A LEARNABLE GRAY-WYNER NETWORK

We now formulate a version of a Gray-Wyner Network that is grounded on the proposed objective function. It separates common and private information between two tasks, as it explores the transmit-receive rate tradeoff. We use learnable entropy models as the rate functions $r_{0,1,2}$ in Eq. 12. These

entropy models produce rates that, as part of the objective function, induce the desired type of information in the representations $Y_{0,1,2}$. Moreover, they allow us to efficiently code (compress) the resulting representations.

Figure 2 provides an overview of the proposed architecture. Inputs $X_1$ and $X_2$ are both processed by two analysis transforms $f_1$ and $f_2$. Because each branch of the proposed architecture has access to both sources $X_1$ and $X_2$, all exclusive information from either source is available to assist in performing tasks $Z_1$ or $Z_2$. This effectively removes the requirement for the conditions in 1.

The output of each analysis transform is passed through a quantization function $q$ that discretizes the representation. It has a differentiable training-time approximation that allows gradient propagation. The representation is then split into two tensors, such that:

$$\left(Y_{\{1,2\}}, Y_0^{(\{1,2\})}\right) = \left(q \circ f_{\{1,2\}}\right)\left(X_{\{1,2\}}\right). \tag{13}$$

Then, $Y_0^{(\{1,2\})}$ are combined into $Y_0$ so that:

$$[Y_0]_i = \begin{cases} 1/2 \left(\left[Y_0^{(1)}\right]_i + \left[Y_0^{(2)}\right]_i\right), & \text{if } \left[Y_0^{(1)}\right]_i = \left[Y_0^{(2)}\right]_i \\ 0, & \text{otherwise,} \end{cases} \tag{14}$$

where $i \in \mathbb{Z}_+$ indexes the elements in the tensors. Using auto-differentiation, the $1/2 \left(Y_0^{(1)} + Y_0^{(2)}\right)$ expression ensures that gradients flow to both inputs wherever elements match. An auxiliary loss term encourages the two input tensors to match. The augmented loss function is given as:

$$\mathcal{L}_{\text{aug}} = \mathcal{L} + \mathbb{E}\left[\frac{\gamma}{|Y_0|}\left\|Y_0^{(1)} - Y_0^{(2)}\right\|_2^2\right], \tag{15}$$

where $\gamma$ influences the impact of this additional loss. Small values of $\gamma$ might result in elements of $Y_0^{(1)}$ and $Y_0^{(2)}$ never matching. A large $\gamma$ can result in degenerate distributions for $Y_0^{(1)}$ and $Y_0^{(2)}$. In both cases, the common channel is underutilized. Thus, this auxiliary loss can discourage the use of the common channel. We overcome this obstacle in practice by setting $\gamma = 1$ and reducing the cost of usage of the common channel $\beta$ when necessary, offering it as the only hyper-parameter.

Entropy models for each private channel, $h_1$ and $h_2$, propose parameters of a probability distribution assumed for its target representations, conditioned on the common representation. This conditioning is prescribed by the proposed objective function. Hence, these entropy models use as context the previously coded elements in $Y_{\{1,2\}}$, in addition to $Y_0$, to predict the distribution parameters of the current elements in $Y_{\{1,2\}}$, respectively. An entropy model for the common channel, $h_0$, only uses as context the previously coded elements in $Y_0$. As we have seen, it is often difficult to discard the information in the common channel from the private channels, such that $I(Y_1, Y_2; Y_0) = 0$. Hence, using the common representation to model the entropy of the private representations can handle the redundancies between the private and common channels and improve compression.

Each private representation is concatenated with the common representation and processed by a synthesis transform for its corresponding task. Each synthesis transform contains a task-specific model to produce reasonable outputs within sample spaces of the targets $Z_{\{1,2\}}$, respectively. Using only the private channel as input for the synthesis transform is another plausible architecture but it forces the common information to be present in the private channel. The conditional entropy models must then be able to predict this common information very accurately to avoid rate increases due to redundancies. This is often difficult for learnable codecs.

Other architectures for the analysis transform $f$ are evaluated in Section 4.1. Intuitively, this architecture provides flexible representations while reducing the learning complexity of making them compatible. A theoretical justification is presented in Appendix C, in which we introduce a measure of compatibility between representations based on the generalization error induced by the hypotheses (family of functions) used to generate them.

# 4 EXPERIMENTAL EVALUATION

In our experiments, the proposed architecture specializes to a single source $X$, so that $(X_1, X_2) = X$. We use an architecture inspired by He et al. (2022) as analysis and synthesis transforms. It

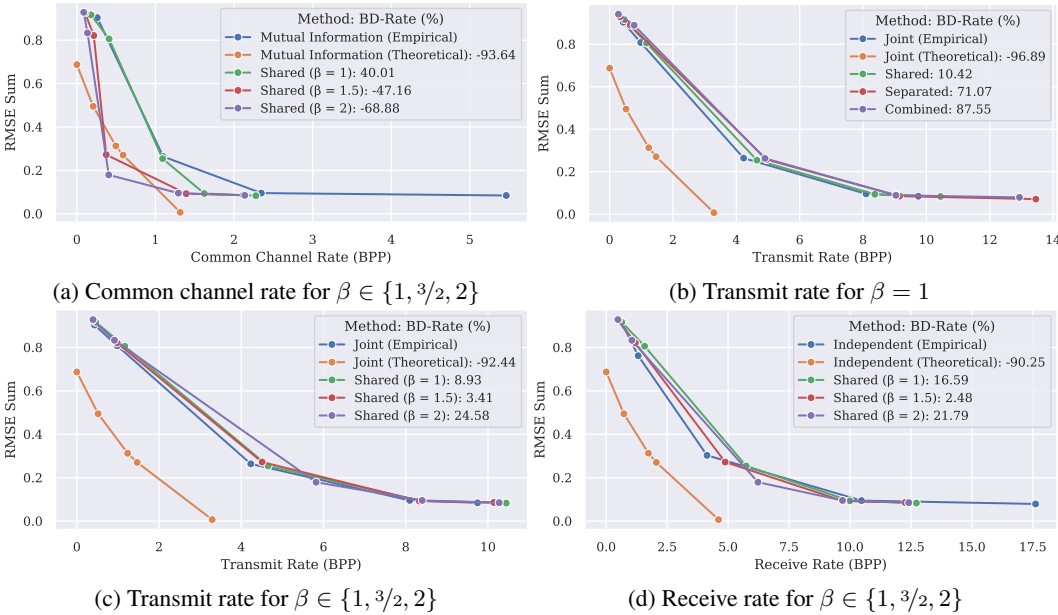

Figure 3: Rate-distortion curves for different methods, optimized for transmit or receive rate, or a mixture of both. BD-rates (Bjontegaard, 2001) are computed with respect to the method with no assigned score. Bits-per-pixel (BPP) is the bitrate scaled by $1/64 \times 64$.

consists of 3 stacks of 3 ResNet blocks each, with the stacks interjecting 4 convolutional layers acting as dimensional bottlenecks. These convolutional layers scale their inputs. As such, the analysis transform progressively reduces the spatial dimensions by a total factor of 16, while increasing the number of channels to 24, 48, 192 and $E$. The synthesis transform performs the opposite in tandem, decreasing the number of channels and increasing the spatial size to that of the original input.

As a quantization function $q$, we use the straight-through gradient function proposed in Theis et al. (2017). We use the relatively simple model of Ballé et al. (2018). The common representation is processed and used in place of the hyper-prior to establish the conditional entropy models. As such, for each private channel, the common representation is processed by 2 stacks of 2 ResNet blocks each, interjected by a convolutional layer that increases the number of channels from $E/2$ to $E$. Masked convolutions turn the entropy model auto-regressive, which is suitable for coding (Ballé et al., 2018).

See Appendix D for architecture diagrams, hyper-parameters, training settings, other results, and further discussions. Code is available at: github.com/adeandrade/research

### 4.1 TRANSMIT-RECEIVE TRADEOFF AND ABLATION STUDY

We developed a synthetic dataset to study the proposed method. Let $X_1$ and $X_2$ index the first dimension of a random variable $X$. We created an $X$ such that $H(X_1, X_2) = 3.3$ bits per element, $H(X_1) + H(X_2) = 4.62$, and consequently, $I(X_1; X_2) = 1.32$. $X_1$ and $X_2$ are individually transformed to generate targets $Z_1$ and $Z_2$ of two linear regression tasks. We train to minimize the RMSE loss between the predictions and the targets of each task.

In addition to the proposed *Shared* architecture, we evaluate the rate-distortion performance of two additional encoder architectures. The *Separated* architecture has an independent analysis transform for each channel. The *Combined* architecture uses a single analysis transform and splits the output tensor into 3 parts corresponding to the GWN channels. If both tasks share a single channel, a resulting *Joint* architecture optimizes the transmit rate. Using a private channel for each task without a common channel, results in an *Independent* method which optimizes the receive rate. The rates produced by these methods are used to compute empirical estimates of the joint and marginal rate-

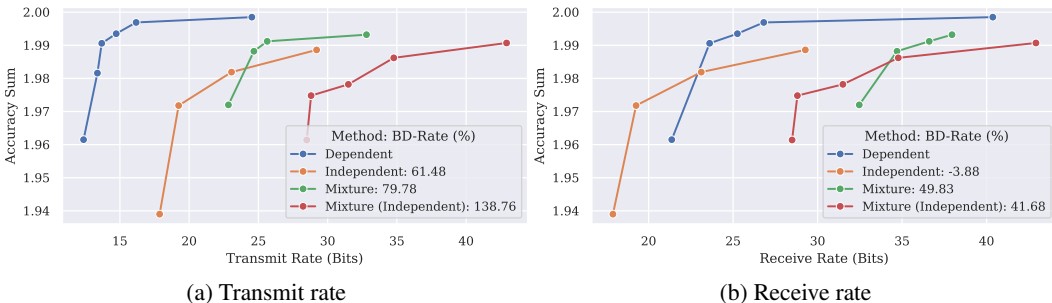

(a) Transmit rate    (b) Receive rate

Figure 4: Validation rate-accuracy curves for colored MNIST on 3 different PMFs. The top-1 classification accuracies are added and shown on the vertical axis. The total bitrate is shown as rate. BD-rates are computed with respect to the Dependent PMF.

distortion functions, and mutual information between tasks. We contrast them against theoretical values. See Appendix D for details on the dataset, architectures, and measurement calculation.

Figure 3a shows that optimizing the transmit rate produces rates for the common channel that are higher than the empirical mutual information between sources. The same figure also shows that optimizing the receive rate produces rates for the common channel that are lower than the empirical mutual information. Codecs trained with $\beta = {}^3/_2$ explore the transmit-receive tradeoff. They have a lower rate on the common channel than the codecs optimized for the transmit rate, but more information than those optimized for the receive rate.

For all $\beta$ explored, the Shared architecture outperforms the Separated and Combined alternatives, as shown in Figure 3b for $\beta = 1$. See Appendix D for results on the other $\beta$. Note that the empirical estimates of the rate are considerably higher than the theoretical values, as often seen in practice (Bajić, 2025). Figures 3c and 3d show that $\beta = {}^3/_2$ is a reasonable value for this problem, since it performs marginally better than $\beta = 1$ and $\beta = 2$, in both transmit and receive rates, respectively.

### 4.2 EDGE-CASE EXPLORATION WITH IMAGE CLASSIFICATION

Classification problems have well-defined theoretical measurements of mutual information against which we can compare. We randomly colorize digit images from MNIST (LeCun et al., 2010) according to three different PMFs: 1. A *Dependent* PMF, where one color always corresponds to one digit; 2. An *Independent* PMF, in which for each sample, one out of 10 colors is sampled uniformly; and 3. A *Mixture* PMF, where each digit has a subset of 10 colors assigned to it, with uniform probability.

One of the tasks in our proposed method predicts the digit, while the other task predicts the color, using the colorized images and corresponding targets. The Dependent PMF has a joint entropy of $\log_2 10$ bits and a mutual information of the same amount. The Independent PMF has a joint entropy of $2 \log_2 10$ bits, and a mutual information of 0 bits. The Mixture PMF has a joint entropy of 5.12 bits, and a mutual information of 1.4 bits.

Figures 4a and 4b show the transmit an receive rates, respectively, for the 3 PMFs. We operate within an order of magnitude of the theoretical bounds, which is comparable to other codecs (Bajić, 2025). More importantly, the model trained on the Dependent PMF produces a lower transmit rate since it places most of its information on the common channel, taking advantage of all information being common. On the other hand, the model trained on the Independent PMF produces the lowest receive rate, since it has a very low rate on the common channel, as with the underlying PMF.

The common information in the Mixture PMF is not very separable, which results in our method producing lower rate-distortion performance compared to the other PMFs. It still performs better, in terms of transmit rate, than the Independent approach from Section 4.1, where there is no common channel. Appendix D shows the rate of each channel, produced by our method, for every PMF.

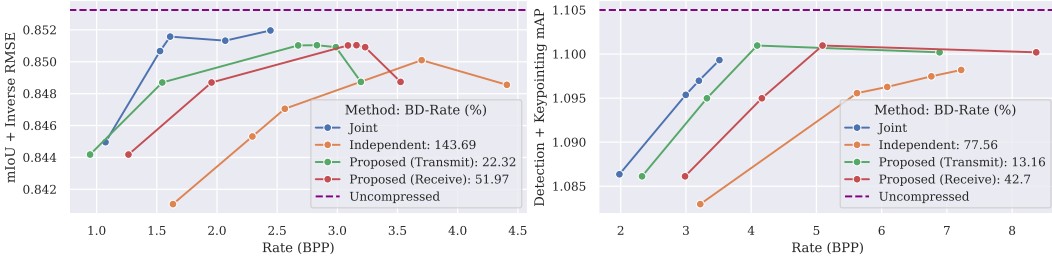

(a) Semantic segmentation and depth estimation    (b) Object detection and keypoint detection

Figure 5: Rate-accuracy curves of the proposed method against the Joint and Independent baselines. The tasks in (a) are reported for the validation set of Cityscapes. The tasks in (b) are reported for the validation set of COCO 2017. The transmit and receive rates of the proposed method are included. BD-rates are computed with respect to the Joint method. The task performances are added. The depth RMSE is scaled so its inverse is in a similar scale as the segmentation mean intersection over union (mIoU). The detection performances are measured by the mean average precision (mAP). The *Uncompressed* lines correspond to the original performances of the pre-trained task models.

### 4.3 RATE-DISTORTION PERFORMANCE IN COMPUTER VISION TASKS

We evaluate the proposed method on semantic segmentation and depth estimation for Cityscapes (Cordts et al., 2016), and on object detection and keypoint detection for COCO 2017 (Lin et al., 2014). As part of $g_1$ and $g_2$, we append pre-trained task-specific models to the synthesis transforms. We keep their weights fixed as we train the rest of the codec. We use DeepLabV3+ with MobileNet for semantic segmentation (Chen et al., 2018a), LRASPP with MobileNetV3 for depth estimation (Howard et al., 2019), Faster R-CNN with ResNet50 for object detection (Ren et al., 2017), and Keypoint R-CNN with ResNet50 for keypoint detection (He et al., 2020).

Figure 5 compares the performance of the proposed method. It is able to outperform an Independent approach and it is relatively close to the Joint approach. The curves for the receive rate are higher than the Independent approach, suggesting that the rate of the common channel is lower than the empirical mutual information between these tasks. Some curves in the Cityscapes experiments have an increase in distortion with the lowest compression, which is often informally attributed to the lack of regularization, provided by stronger rate constraints.

## 5 SUMMARY AND CONCLUSION

We validated the ability of the proposed learnable Gray-Wyner Network to distill common information between tasks and compared it against other architectures. The performance is theoretically justified by analyzing the compatibility between intermediate representations. We provided bounds that relate two measures of lossy common information. The proposed optimization objective is derived from this theory and was able to empirically explore the transmit-receive rate tradeoff. The proposed method is also able to handle edge-cases, including a case where there is no mutual information between tasks, and another where the tasks are fully dependent. Finally, between the three computer vision experiments, our codecs achieved, on average, a BD-rate advantage of -81.58% in transmit rate, against single-task codecs.

Isolating and coding the common information between dependent tasks allows for the efficient distributed inference of machine tasks. Generating representations that explore the tradeoff between the transmit and receive rates in the Gray-Wyner Network has additional practical implications in storage and selective retrieval, and dispersive information routing (Viswanatha et al., 2011). Knowing the information requirements of learned representations can assist in planning for the resources allocated to a neural network, its dimensionality, and quantization levels.

Extensions to three or more tasks are possible, but since the total number of channels scales exponentially, a more dynamic architecture might be required. Nevertheless, the theoretical contributions of this work should prove useful in deriving new methods.

ACKNOWLEDGMENTS

This work was partially funded by Intel Labs and the Natural Sciences and Engineering Research Council of Canada (NSERC).

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

# A   BOUNDS FOR LOSSY COMMON INFORMATION

## A.1   ADDITIONAL PRELIMINARIES

Recall that the concept of mutual information measures the mutual dependence between two random variables. For two discrete distributions, it can be given as:

$$I(X_1; X_2) = \sum_{x_1 \in \mathcal{X}_1} \sum_{x_2 \in \mathcal{X}_2} P_{X_1, X_2}(x_1, x_2) \log \frac{P_{X_1, X_2}(x_1, x_2)}{P_{X_1}(x_1) P_{X_2}(x_2)}. \tag{16}$$

The concept also applies to continuous random variables, where a summation can be replaced with an integral over the corresponding sample space. The variables $X_1$ or $X_2$ can be extended to joint probability functions of two or more variables, supporting concepts such as:

$$I(X_{1:n}; Y_{1:n}) = \sum_{x_{1:n} \in \mathcal{X}_{1:n}} \sum_{y_{1:n} \in \mathcal{Y}_{1:n}} P_{X_{1:n}, Y_{1:n}}(x_{1:n}, y_{1:n}) \log \frac{P_{X_{1:n}, Y_{1:n}}(x_{1:n}, y_{1:n})}{P_{X_{1:n}}(x_{1:n}) P_{Y_{1:n}}(y_{1:n})}, \tag{17}$$

where $X_{1:n} = X_1, ..., X_n$ and $Y_{1:n} = Y_1, ..., Y_n$ are all random variables. The conditional mutual information measure is the expected value of the mutual information of two random variables given the value of a third. It can be given as:

$$I(X; Y|Z) = \sum_{z \in \mathcal{Z}} \sum_{y \in \mathcal{Y}} \sum_{x \in \mathcal{X}} P_{X, Y, Z}(x, y, z) \log \frac{P_{X, Y|Z}(x, y|z)}{P_{X|Z}(x|z) P_{Y|Z}(y|z)}, \tag{18}$$

where $Z$ is any discrete random variable. Similarly to mutual information, conditional mutual information is non-negative and can be extended to continuous random variables and joint probability distributions of two of more random variables.

The joint rate-distortion is given as:

$$R_{X_1, X_2}(D_1, D_2) = \min I(X_1, X_2; \hat{Z}_1, \hat{Z}_2), \tag{19}$$

where the minimum is taken with respect to all tuples $(\hat{Z}_1, \hat{Z}_2)$ that achieve at least $(D_1, D_2)$ distortion. The marginal rate-distortion functions are given by:

$$R_{X_1}(D_1) = \min I(X_1; \hat{Z}_1), \qquad R_{X_2}(D_2) = \min I(X_2; \hat{Z}_2), \tag{20}$$

with similar distortion restrictions on $\hat{Z}_1$ and $\hat{Z}_2$. The conditional rate-distortion function is:

$$R_{X_1|X_2}(D_1) = \min I(X_1; \hat{Z}_1|X_2), \qquad R_{X_2|X_1}(D_2) = \min I(X_2; \hat{Z}_2|X_1), \tag{21}$$

with the corresponding distortion restrictions on the minimums.

Recall that the *interaction information (II)* (Ting, 1962; Yeung, 1991) between random variables $X$, $Y$, and $Z$ is given as:

$$I(X; Y; Z) = I(X; Y) - I(X; Y|Z), \tag{22}$$

and that the chain rule of mutual information (Cover & Thomas, 2006) is defined as:

$$I(X; Z) = I(X; Y, Z) - I(X; Y|Z). \tag{23}$$

We derive the following corollary:

$$I(X; Y; Z) = I(X, Y; Z) - I(X; Z|Y) - I(Y; Z|X). \tag{24}$$

An outline of the proof is presented as follows:

$$I(X; Y; Z) = \tfrac{1}{2} \left[ I(X; Z) - I(X; Z|Y) + I(Y; Z) - I(Y; Z|X) \right] \qquad \text{(Eq. 22 twice)} \tag{25}$$
$$= I(X, Y; Z) - I(X; Z|Y) - I(Y; Z|X). \qquad \text{(Eq. 23 twice)} \tag{26}$$

The chain rule of mutual information and the definition of interaction information can be extended to any finite amount of random variables. Interaction information can be negative when the variables involved are conditional dependent.

### A.2 SUPPORTING STATEMENTS

**Lemma 1.** *Let $\hat{\mathcal{Z}}_{D_1,D_2}^{(t)}$ be the set of tuples $(\hat{Z}_1, \hat{Z}_2)$ that achieve $R_{X_1,X_2}(D_1, D_2)$, and $\hat{\mathcal{Z}}_{D_1,D_2}^{(r)}$ be the set of tuples $(\hat{Z}_1, \hat{Z}_2)$ such that $\hat{Z}_1$ achieves $R_{X_1}(D_1)$, and $\hat{Z}_2$ achieves $R_{X_2}(D_2)$. We have that:*

$$\max_{(\hat{Z}_1,\hat{Z}_2)\in\hat{\mathcal{Z}}_{D_1,D_2}^{(r)}} I(X_1, X_2; \hat{Z}_1; \hat{Z}_2) \leq \min_{(\hat{Z}_1,\hat{Z}_2)\in\hat{\mathcal{Z}}_{D_1,D_2}^{(t)}} I(X_1, X_2; \hat{Z}_1; \hat{Z}_2). \tag{27}$$

*Proof.* By definition, we have:

$$I\left(X_1; \hat{Z}_1^{(r,1)}\right) + I\left(X_2; \hat{Z}_2^{(r,1)}\right) = I\left(X_1; \hat{Z}_1^{(r,2)}\right) + I\left(X_2; \hat{Z}_2^{(r,2)}\right) \tag{28}$$

$$\forall \left(\hat{Z}_1^{(r,1)}, \hat{Z}_2^{(r,1)}\right), \left(\hat{Z}_1^{(r,2)}, \hat{Z}_2^{(r,2)}\right) \in \hat{\mathcal{Z}}_{D_1,D_2}^{(r)}. \tag{29}$$

The minimization objective for $R_{X_1}(D_1) + R_{X_2}(D_2)$, over all possible tuples $(\hat{Z}_1, \hat{Z}_2)$, is given as:

$$I(X_1; \hat{Z}_1) + I(X_2; \hat{Z}_2). \tag{30}$$

This objective selects $\hat{Z}_1$ and $\hat{Z}_2$ independently and does not consider their mutual information. Thus, solutions with lower or higher mutual information are not penalized and are present in $\hat{\mathcal{Z}}_{D_1,D_2}^{(r)}$, as long as they meet the objective. Moreover, the objective does not allow $\hat{Z}_1$ or $\hat{Z}_2$ to have excess information in $I(X_1; \hat{Z}_1)$ or $I(X_2; \hat{Z}_2)$ to achieve $D_1$ and $D_2$, respectively, since a different tuple would exist that discards this information, setting a new goal for the accepted solutions in $\hat{\mathcal{Z}}_{D_1,D_2}^{(r)}$.

To simplify notation, let $X = (X_1, X_2)$. By definition, we have:

$$I\left(X; \hat{Z}_1^{(t,1)}, \hat{Z}_2^{(t,1)}\right) = I\left(X; \hat{Z}_1^{(t,2)}, \hat{Z}_2^{(t,2)}\right) \forall \left(\hat{Z}_1^{(t,1)}, \hat{Z}_2^{(t,1)}\right), \left(\hat{Z}_1^{(t,2)}, \hat{Z}_2^{(t,2)}\right) \in \hat{\mathcal{Z}}_{D_1,D_2}^{(t)} \tag{31}$$

The minimization objective in $R_{X_1,X_2}(D_1, D_2)$, over all possible tuples $(\hat{Z}_1, \hat{Z}_2)$, is broken down as:

$$I(X; \hat{Z}) \tag{32}$$

$$= I(X; \hat{Z}_1) + I(X; \hat{Z}_2|\hat{Z}_1) \tag{Eq. 23} \tag{33}$$

$$= \underbrace{I(X; \hat{Z}_1)}_{(a)} + \underbrace{I(X; \hat{Z}_2)}_{(b)} - \underbrace{I(X; \hat{Z}_1; \hat{Z}_2)}_{(c)} \tag{Eq. 22} \tag{34}$$

$$= \underbrace{I(X_1; \hat{Z}_1)}_{(a.1)} + \underbrace{I(X_2; \hat{Z}_1|X_1)}_{(a.2)} + \underbrace{I(X_2; \hat{Z}_2)}_{(b.1)} + \underbrace{I(X_1; \hat{Z}_2|X_2)}_{(b.2)} - \underbrace{I(X; \hat{Z}_1; \hat{Z}_2)}_{(c)}. \tag{Eq. 23} \tag{35}$$

Remarks:

1. Because of term (c), this objective considers the common information between variables, at the expense of allowing tuples to have private information from the other variable in them.

2. Information in (a.1) and (b.1) contributes to achieving $D_1$ and $D_2$, respectively.

3. Information in (a.2) and (b.2) does not contribute to achieving $D_1$ or $D_2$, respectively.

4. Interaction information $I(X; \hat{Z}_1; \hat{Z}_2)$ is present in the same proportion in (a) and (b), each.

5. If some interaction information $I(X; \hat{Z}_1; \hat{Z}_2)$ is not in (a.1), it is in (a.2). If it is not in (b.1), it must be in (b.2).

6. Interaction information $I(X; \hat{Z}_1; \hat{Z}_2)$ is distributed among (a.1) and (b.1), (a.1) and (b.2), and (a.2) and (b.1).

7. Interaction information $I(X; \hat{Z}_1; \hat{Z}_2)$ present in (a.2) and (b.2) has no purpose and the minimization objective would select tuples where it does not exist, since the penalty is double and is not sufficiently countered by (c), increasing the objective.

8. Since information in $\hat{Z}_1$ of type (a.2) or information in $\hat{Z}_2$ of type (b.2) does not contribute to achieving $D_1$ or $D_2$, respectively, the minimization objective would select tuples that have zero information of those types, *unless* it is also present in the other variable ($\hat{Z}_2$ or $\hat{Z}_1$), so that it is countered by (c) without affecting the objective.

9. Interaction information $I(X; \hat{Z}_1; \hat{Z}_2)$ that reduces the private information in (a.1) and (b.1), takes precedence over other information when it decreases the objective.

Since all elements in $\hat{\mathcal{Z}}^{(r)}_{D_1,D_2}$ have no type (a.2) or (b.2) information and all have the same amount of type (a.1) and (b.1) information, only the term (c) differentiates them under the $R_{X_1,X_2}$ objective. Hence, the tuples with the most common information will have the lowest objective:

$$I\left(X; \hat{Z}^{(r_{\max})}\right) \leq I\left(X; \hat{Z}_1^{(r_{\min})}, \hat{Z}_2^{(r_{\min})}\right), \tag{36}$$

where $\hat{Z}^{(r_{\max})}$ and $\hat{Z}^{(r_{\min})}$ are tuples in $\hat{\mathcal{Z}}^{(r)}_{D_1,D_2}$ with the highest and lowest $I(X; \hat{Z}_1; \hat{Z}_2)$, respectively.

Let $\hat{Z}^{(t_{\min})}$ be the tuple in $\hat{\mathcal{Z}}^{(t)}_{D_1,D_2}$ with the lowest $I(X; \hat{Z}_1; \hat{Z}_2)$. This tuple has no information of type (a.2) or (b.2), since it does not contribute to reducing the objective. Moreover, this property, in conjunction with the conditions in 1, establishes that $\hat{Z}_1^{(t_{\min})}$ and $\hat{Z}_2^{(t_{\min})}$ can be represented as functions of $X_1$ and $X_2$, respectively. Since these functions are applied independently to different variables, the resulting variables cannot be conditionally dependent on $X$. Thus, the interaction information involving these variables is non-negative. Using these properties, we have:

$$I(X; \hat{Z}) \tag{37}$$

$$= I(X_1; \hat{Z}_1) + I(X_2; \hat{Z}_1|X_1) + I(X_2; \hat{Z}_2) + I(X_1; \hat{Z}_2|X_2) - I(X; \hat{Z}_1; \hat{Z}_2) \quad \text{(Eq. 35)} \tag{38}$$

$$= I\left(X_1; \hat{Z}_1^{(t_{\min})}\right) + I\left(X_2; \hat{Z}_2^{(t_{\min})}\right) - I\left(X; \hat{Z}_1^{(t_{\min})}; \hat{Z}_2^{(t_{\min})}\right) \quad \text{(no a.2/b.2)} \tag{39}$$

$$\leq I\left(X_1; \hat{Z}_1^{(t_{\min})}\right) + I\left(X_2; \hat{Z}_2^{(t_{\min})}\right). \quad (\text{II} \geq 0) \tag{40}$$

Thus, tuples that result from optimizing $R_{X_1,X_2}(D_1, D_2)$ can produce lower values than tuples resulting from optimizing $R_{X_1}(D_1) + R_{X_2}(D_2)$, such that:

$$I\left(X; \hat{Z}^{(t_{\min})}\right) \leq I\left(X; \hat{Z}^{(r_{\max})}\right). \tag{41}$$

A minor increase in private information at the expense of greater common information in (a.1) and (b.1) can cause the objective to decrease further than any tuple optimized for $R_{X_1}(D_1) + R_{X_2}(D_2)$. This result conforms to the one shown by Gray (1973), stating that:

$$R_{X_1,X_2}(D_1, D_2) \leq R_{X_1}(D_1) + R_{X_2}(D_2). \tag{42}$$

$\square$

## A.3 MAIN THEOREM

**Theorem 1.** *Let $\hat{\mathcal{Z}}^{(t)}_{D_1,D_2}$ be the set of tuples $(\hat{Z}_1, \hat{Z}_2)$ that achieve $R_{X_1,X_2}(D_1, D_2)$, and $\hat{\mathcal{Z}}^{(r)}_{D_1,D_2}$ be the set of tuples $(\hat{Z}_1, \hat{Z}_2)$, such that $\hat{Z}_1$ achieves $R_{X_1}(D_1)$, and $\hat{Z}_2$ achieves $R_{X_2}(D_2)$. Then:*

$$K(X_1, X_2; D_1, D_2) \leq \max_{(\hat{Z}_1, \hat{Z}_2) \in \hat{\mathcal{Z}}^{(r)}_{D_1,D_2}} I(X_1, X_2; \hat{Z}_1; \hat{Z}_2) \tag{6}$$

$$\leq \min_{(\hat{Z}_1, \hat{Z}_2) \in \hat{\mathcal{Z}}^{(t)}_{D_1,D_2}} I(X_1, X_2; \hat{Z}_1; \hat{Z}_2) \leq C(X_1, X_2; D_1, D_2). \tag{7}$$

*We have equality everywhere iff the maximum and minimum coincide at $(\hat{Z}_1^*, \hat{Z}_2^*)$, and we can represent $\hat{Z}_1^*$ as $(\hat{Z}_1', W)$ and $\hat{Z}_2^*$ as $(\hat{Z}_2', W)$ such that Conditions 3 and 5 hold for $\hat{Z}_1^*, \hat{Z}_2^*$, and $W$.*

*Proof.* We reinstate the conditions for Wyner's common information individually:

$$\hat{Z}_1 \leftrightarrow U \leftrightarrow \hat{Z}_2, \tag{43}$$

$$(X_1, X_2) \leftrightarrow (\hat{Z}_1, \hat{Z}_2) \leftrightarrow U, \tag{44}$$

and do the same for Gács-Körner common information:

$$X_2 \leftrightarrow X_1 \leftrightarrow V, \tag{45}$$

$$X_1 \leftrightarrow X_2 \leftrightarrow V, \tag{46}$$

$$X_1 \leftrightarrow \hat{Z}_1 \leftrightarrow V, \tag{47}$$

$$X_2 \leftrightarrow \hat{Z}_2 \leftrightarrow V. \tag{48}$$

To simplify notation, let $X = (X_1, X_2)$, and $\hat{Z} = (\hat{Z}_1, \hat{Z}_2)$. We show for the RHS that:

$$I(X; \hat{Z}_1; \hat{Z}_2) = I(X; \hat{Z}_1; \hat{Z}_2; U) + I(X; \hat{Z}_1; \hat{Z}_2|U) \qquad \text{(Eq. 22)} \quad (49)$$

$$= I(X; \hat{Z}_1; \hat{Z}_2; U) \qquad \text{(Cond. 43)} \quad (50)$$

$$= I(X; \hat{Z}; U) - I(X; \hat{Z}_1; U|\hat{Z}_2) - I(X; \hat{Z}_2; U|\hat{Z}_1) \qquad \text{(Eq. 24)} \quad (51)$$

$$\leq I(X; \hat{Z}; U) \qquad (I(X; W; Y|Z) \geq 0) \quad (52)$$

$$= I(X; U) - I(X; U|\hat{Z}) \qquad \text{(Eq. 22)} \quad (53)$$

$$= I(X; U). \qquad \text{(Cond. 44)} \quad (54)$$

Thus:

$$I(X; \hat{Z}_1; \hat{Z}_2) \leq I(X; U) \implies \inf I(X; \hat{Z}_1; \hat{Z}_2) \leq \inf I(X; U) \qquad (55)$$

$$\implies \min I(X; \hat{Z}_1; \hat{Z}_2) \leq C(X_1, X_2; D_1, D_2), \quad \text{(Lemma 1)} \quad (56)$$

where the infimum is over the same set as in Eq. 2, and the minimum is over the same set as in Lemma 1.

To prove the LHS, first we derive the following:

$$I(X; \hat{Z}_1; V|\hat{Z}_2) = I(X_1; \hat{Z}_1; V|\hat{Z}_2, X_2) + I(X_2; \hat{Z}_1; V|\hat{Z}_2) = 0, \qquad \text{(Eq. 23, 46, 48)} \quad (57)$$

$$I(X; \hat{Z}_2; V|\hat{Z}_1) = I(X_2; \hat{Z}_2; V|\hat{Z}_1, X_1) + I(X_1; \hat{Z}_2; V|\hat{Z}_1) = 0, \qquad \text{(Eq. 23, 45, 47)} \quad (58)$$

$$I(X; V|\hat{Z}) = I(X_2; V|\hat{Z}) + I(X_1; V|\hat{Z}, X_2) = 0. \qquad \text{(Eq. 23, 48, 46)} \quad (59)$$

Now, we have:

$$I(X; \hat{Z}_1; \hat{Z}_2) = I(X; \hat{Z}_1; \hat{Z}_2; V) + I(X; \hat{Z}_1; \hat{Z}_2|V) \qquad \text{(Eq. 22)} \quad (60)$$

$$\geq I(X; \hat{Z}_1; \hat{Z}_2; V) \qquad (I(X; W; Y|Z) \geq 0) \quad (61)$$

$$= I(X; \hat{Z}; V) - I(X; \hat{Z}_1; V|\hat{Z}_2) - I(X; \hat{Z}_2; V|\hat{Z}_1) \qquad \text{(Eq. 24)} \quad (62)$$

$$= I(X; \hat{Z}; V) \qquad \text{(Eq. 57,58)} \quad (63)$$

$$= I(X; V) - I(X; V|\hat{Z}) \qquad \text{(Eq. 22)} \quad (64)$$

$$= I(X; V). \qquad \text{(Eq. 59)} \quad (65)$$

Finally, we have:

$$I(X; V) \leq I(X; \hat{Z}_1; \hat{Z}_2) \implies \sup I(X; V) \leq \sup I(X; \hat{Z}_1; \hat{Z}_2) \qquad (66)$$

$$\implies K(X_1, X_2; D_1, D_2) \leq \max I(X; \hat{Z}_1; \hat{Z}_2), \quad \text{(Lemma 1)} \quad (67)$$

where the supremum is over the same set as in Eq. 4, and the maximum is over the same set as in Lemma 1.

To obtain equality on the RHS, Equations 57 and 58 shows us under which conditions the last two terms in Equation 51 are zero. Since the conditions hold in equality for $\hat{Z}^*$ and $W$, we continue the same analysis with the corresponding terms set to zero, yielding:

$$I(X; \hat{Z}_1; \hat{Z}_2; W) = I(X; \hat{Z}; W) \implies I(X; \hat{Z}_1; \hat{Z}_2) = I(X; W) \qquad \text{(Cds. 45-48)} \quad (68)$$

$$\implies \inf I(X; \hat{Z}_1; \hat{Z}_2) = \inf I(X; W) \qquad (69)$$

$$\implies I(X; \hat{Z}_1^*; \hat{Z}_2^*) = C(X_1, X_2; D_1, D_2), \quad \text{(Lemma 1)} \quad (70)$$

where the infimum is over the same conditions as in Eq. 2.

For the LHS, due to Condition 43 holding in equality for $\hat{Z}^*$ and $W$, we have equality in Equation 61. Hence:

$$I(X; \hat{Z}_1; \hat{Z}_2) = I(X; \hat{Z}_1; \hat{Z}_2; W) \implies I(X; W) = I(X; \hat{Z}_1; \hat{Z}_2) \qquad \text{(Cond. 43)} \quad (71)$$

$$\implies \sup I(X; W) = \sup I(X; \hat{Z}_1; \hat{Z}_2) \qquad (72)$$

$$\implies K(X_1, X_2; D_1, D_2) = I(X; \hat{Z}_1^*; \hat{Z}_2^*), \quad \text{(Lemma 1)} \quad (73)$$

where the supremum is over the same conditions as in Eq. 4.

If either of Conditions 45-48 is not met, more than $I(X; \hat{Z}_1^*; \hat{Z}_1^*)$ is present in $W$ and equality is not achieved in Equation 52. If Condition 43 is not met, not all mutual information between $\hat{Z}_1^*$ and $\hat{Z}_2^*$, in common with $(X_1, X_2)$, is placed on $W$ and equality is not achieved in Equation 61. If all these conditions are met, but we have that:

$$\max_{(\hat{Z}_1,\hat{Z}_2)\in\hat{\mathcal{Z}}_{D_1,D_2}^{(r)}} I(X_1, X_2; \hat{Z}_1; \hat{Z}_2) \leq \min_{(\hat{Z}_1,\hat{Z}_2)\in\hat{\mathcal{Z}}_{D_1,D_2}^{(t)}} I(X_1, X_2; \hat{Z}_1; \hat{Z}_2), \tag{74}$$

then the optimization of the joint rate-distortion and the marginal rate-distortion objectives produce tuples with different amounts of mutual information, and the lossy common information measures will not match. $\qquad\square$

## B   DERIVATION OF THE OPTIMIZATION OBJECTIVE

**Theorem 2.** *Assume that $Y_0 = f_0(X_1, X_2); f_0 \in \mathcal{F}_0, Y_1 = f_1(X_1); f_1 \in \mathcal{F}_1, Y_2 = f_2(X_2); f_2 \in \mathcal{F}_2$ are all deterministic functions of their corresponding inputs, and that $g_1 \in \mathcal{G}_1$ and $g_2 \in \mathcal{G}_2$, where $\mathcal{F}_{\{0,1,2\}}$ and $\mathcal{G}_{\{1,2\}}$ are families of functions such that there exits a $f_0, f_1, f_2, g_1,$ and $g_2$ in their respective families that achieve $T(\alpha_1, \alpha_2; D_1, D_2)$. Then:*

$$T(\alpha_1, \alpha_2; D_1, D_2) = \inf\left\{H(Y_0) + \alpha_1 H(Y_1|Y_0) + \alpha_2 H(Y_2|Y_0)\right\}, \tag{10}$$

*where $H(\cdot)$ is Shannon's entropy function, and $H(\cdot|\cdot)$ is the conditional entropy function (Cover & Thomas, 2006). The infimum is over all $f_0, f_1, f_2, g_1,$ and $g_2$ in their corresponding family of functions, such that we obtain, at most, distortions $D_1$ and $D_2$.*

*Proof.* Since $Y_0$ is a deterministic function of $(X_1, X_2)$, we have:

$$I(X_1, X_2; Y_0) \triangleq H(Y_0) - H(Y_0|X_1, X_2) = H(Y_0). \tag{75}$$

Since $Y_1$ is a deterministic function of $X_1$, we have for the conditional rate-distortion function that:

$$R_{X_1|Y_0}(D_1) \triangleq \min I(X_1; Y_1|Y_0) = \min\left\{H(Y_1|Y_0) - H(Y_1|X_1, Y_0)\right\} = \min H(Y_1|Y_0), \tag{76}$$

where the minimum is over all probability distributions $P(Y_1|X_1) \in \mathcal{P}_{Y_1|X_1}$ such that the resulting $Y_1$ produces at most distortion $D_1$. The converse holds, where, since $Y_2$ is a deterministic function of $X_2$, we have that:

$$R_{X_2|Y_0}(D_2) \triangleq \min I(X_2; Y_2|Y_0) = \min\left\{H(Y_2|Y_0) - H(Y_2|X_2, Y_0)\right\} = \min H(Y_2|Y_0), \tag{77}$$

where the minimum is over all probability distributions $P(Y_2|X_2) \in \mathcal{P}_{Y_2|X_2}$ such that the resulting $Y_2$ produces at most distortion $D_2$.

An optimization over probability distributions of random variables can be swapped by an optimization over a family of functions generating those random variables, as long as the function producing the probability distribution minimizing the first problem is in the family of functions. Under this assumption, replacing the terms in Eq. 9:

$$T(\alpha_1, \alpha_2; D_1, D_2) \triangleq \inf\left\{I(X_1, X_2; Y_0) + \alpha_1 R_{X_1|Y_0}(D_1) + \alpha_2 R_{X_2|Y_0}(D_2)\right\}, \tag{78}$$

with Equations 75, 76, and 77, we arrive at the desired result. $\qquad\square$

## C   THEORY OF COMPATIBLE REPRESENTATIONS

We define the compatibility between two representations as the measurable difficulty of learning a task that tries to match them in Euclidean space. Since their information content can be different, it might not be possible for the two representations to closely match. However, we do not propose to use as a measure the loss itself but the generalization error induced by the family of functions used to generate these representations. An increase in the upper-bound on this generalization error means that the corresponding model has the potential to make the representations more difficult to bring back to a common manifold.

Within this section, we override all previous established notation:

**Definition 1.** *Let $f : \mathbb{R}^D \to \mathbb{R}^D$ be a function representing a neural network layer such as as dense, convolutional, ResNet, or transformer layer. Let $\sigma$ be an element-wise function, such as a ReLU, ELU, sigmoid, tanh, or an identity function. Let $f \in \mathcal{F}$ be a family of neural network layers. We define a neural network family of $L \in \mathbb{Z}_+$ homogeneous layers as:*

$$\mathcal{F}_L = \big\{ f : f(\mathbf{y}) = (f_L \circ ... \circ \sigma \circ f_2 \circ \sigma \circ f_1)(\mathbf{y}), \; \forall (f_1, ..., f_L) \in \mathcal{F}_1 \times ... \times \mathcal{F}_L; \tag{79}$$

$$\mathcal{F}_i = \mathcal{F}, \; \forall i \in \{1, ..., L\} \big\}. \tag{80}$$

*Let $\mathcal{D} = \{([\mathbf{y_1}]_i, [\mathbf{y_2}]_i)\}_{i=1}^N \sim Y_1, Y_2$ be a set of samples from the joint distribution $Y_1, Y_2$. We define the compatibility between representations $Y_1$ and $Y_2$ as the generalization error $C_{\mathcal{F}_L}(\mathcal{D}; Y_1, Y_2)$ of a regression task of $Y_1$ onto $Y_2$:*

$$C_{\mathcal{F}_L}(\mathcal{D}; Y_1, Y_2) = \sup_{\ell \in \mathcal{L}_{\mathcal{F}_L}} |E_{Y_1, Y_2}(\ell) - E_{\mathcal{D}}(\ell)|; \tag{81}$$

$$\mathcal{L}_{\mathcal{F}_L} = \{\ell : \ell(\mathbf{y}_1, \mathbf{y}_2) = \|f(\mathbf{y}_1) - \mathbf{y}_2\|, \; \forall f \in \mathcal{F}_L\}, \tag{82}$$

$$E_{Y_1, Y_2}(\ell) = \mathbb{E}_{(\mathbf{y}_1, \mathbf{y}_2) \sim Y_1, Y_2}[\ell(\mathbf{y}_1, \mathbf{y}_2)], \; E_{\mathcal{D}}(\ell) = \frac{1}{N} \sum_{(\mathbf{y}_1, \mathbf{y}_2) \in \mathcal{D}} \ell(\mathbf{y}_1, \mathbf{y}_2). \tag{83}$$

**Lemma 2.** *Let the functions in the family $\mathcal{S} \subset \mathcal{F}_L$ be Lipschitz such that $\mathrm{Lip}(f) \le \mu \; \forall f \in \mathcal{S}$, where $\mathrm{Lip}(f)$ retrieves the Lipschitz constant of a function $f$. The Rademacher complexity $\mathrm{Rad}_{\mathcal{D}}(\mathcal{L}_{\mathcal{S}})$ is bounded by:*

$$\mathrm{Rad}_{\mathcal{D}}(\mathcal{L}_{\mathcal{S}}) \le \mu \mathrm{Rad}(\mathcal{D}_1) + \mathrm{Rad}(\mathcal{D}_2), \tag{84}$$

*where $\mathcal{D}_1 = \{\mathbf{y}_1 : (\mathbf{y}_1, \mathbf{y}_2) \in \mathcal{D}\}$, $\mathcal{D}_2 = \{\mathbf{y}_2 : (\mathbf{y}_1, \mathbf{y}_2) \in \mathcal{D}\}$, and $\mathrm{Rad}(\mathcal{D})$ is the Rademacher complexity of a set $\mathcal{D}$.*

*Proof.* Let $A$ be a random variable with sample space $\{-1, 1\}^N$ following the Rademacher distribution. We have that:

$$\mathrm{Rad}_{\mathcal{D}}(\mathcal{L}_{\mathcal{S}}) = \frac{1}{N} \mathbb{E}_{\mathbf{a} \sim A} \left[ \sup_{\ell \in \mathcal{L}_{\mathcal{S}}} \left| \sum_{i=1}^N a_i \ell([\mathbf{y}_1]_i, [\mathbf{y}_2]_i) \right| \right] \tag{85}$$

$$\le \frac{1}{N} \mathbb{E}_{\mathbf{a} \sim A} \left[ \sup_{f \in \mathcal{S}} \left| \sum_{i=1}^N a_i [f([\mathbf{y}_1]_i) - [\mathbf{y}_2]_i] \right| \right] \tag{86}$$

$$= \frac{1}{N} \mathbb{E}_{\mathbf{a} \sim A} \left[ \sup_{f \in \mathcal{S}} \left| \sum_{i=1}^N a_i f([\mathbf{y}_1]_i) - \sum_{i=1}^N a_i [\mathbf{y}_2]_i \right| \right] \tag{87}$$

$$\le \frac{1}{N} \mathbb{E}_{\mathbf{a} \sim A} \left[ \sup_{f \in \mathcal{S}} \left\{ \left| \sum_{i=1}^N a_i f([\mathbf{y}_1]_i) \right| + \left| \sum_{i=1}^N a_i [\mathbf{y}_2]_i \right| \right\} \right] \tag{88}$$

$$= \frac{1}{N} \mathbb{E}_{\mathbf{a} \sim A} \left[ \sup_{f \in \mathcal{S}} \left| \sum_{i=1}^N a_i f([\mathbf{y}_1]_i) \right| \right] + \frac{1}{N} \mathbb{E}_{\mathbf{a} \sim A} \left[ \sup_{f \in \mathcal{S}} \left| \sum_{i=1}^N a_i [\mathbf{y}_2]_i \right| \right] \tag{89}$$

$$= \mathrm{Rad}_{\mathcal{D}_1}(\mathcal{S}) + \mathrm{Rad}(\mathcal{D}_2) \tag{90}$$

$$\le \mu \mathrm{Rad}(\mathcal{D}_1) + \mathrm{Rad}(\mathcal{D}_2). \tag{91}$$

Eq. 86 uses the Kakade & Tewari Lemma (Kakade & Tewari, 2008) based on Talagrand's contraction principle (Ledoux & Talagrand, 2013; Bartlett & Mendelson, 2002). It states that if all vectors in a set $A$ are operated by a Lipschitz function, then $\mathrm{Rad}(A)$ is at most multiplied by the Lipschitz constant of the function. It is easy to show that the norm is 1-Lipschitz. Eq. 91 uses the same lemma. Eq. 88 uses $|a - b| \le |a| + |b|$. $\qquad \square$

We can upper-bound the compatibility measure $C_{\mathcal{S}}(\mathcal{D}; Y_1, Y_2)$ using a well-known generalization error bound in terms of Rademacher complexity:

**Theorem 3.** *Let the functions in the family $\mathcal{S} \subset \mathcal{F}_L$ be Lipschitz such that $\mathrm{Lip}(f) \le \mu \; \forall f \in \mathcal{S}$, and let the loss function be bounded such that $\ell(\mathbf{y}_1, \mathbf{y}_2) \in [-B, B] \; \forall \ell \in \mathcal{L}_{\mathcal{S}}, \mathbf{y}_1 \in \mathcal{Y}_1, \mathbf{y}_2 \in \mathcal{Y}_2$, where $\mathcal{Y}_1$ and $\mathcal{Y}_2$ are the sample spaces of $Y_1$ and $Y_2$, respectively. Then, $\forall \delta \in (0, 1)$, with probability at least $1 - \delta$, we have:*

$$C_{\mathcal{S}}(\mathcal{D}; Y_1, Y_2) \le 2[\mu \mathrm{Rad}(\mathcal{D}_1) + \mathrm{Rad}(\mathcal{D}_2)] + B\sqrt{2/N \log 2/\delta} \tag{92}$$

*Proof.*

$$C_{\mathcal{S}}(\mathcal{D}; Y_1, Y_2) \leq 2\mathrm{Rad}_{\mathcal{D}}(\mathcal{L}_{\mathcal{S}}) + B\sqrt{2/N \log 2/\delta} \tag{93}$$

$$\leq 2\left[\mu\mathrm{Rad}(\mathcal{D}_1) + \mathrm{Rad}(\mathcal{D}_2)\right] + B\sqrt{2/N \log 2/\delta}. \tag{94}$$

Eq. 93 uses Theorem 26.5 in Shalev-Shwartz & Ben-David (2014). Eq. 94 uses Lemma 2. □

**Theorem 4.** *Assume that the representations $Y_1$ and $Y_2$ are functions of a common random variable $X$, such that:*

$$Y_1 = g_1(X); g_1 \in \mathcal{G}_{L_1}; \mathcal{G}_{L_1} \subset \mathcal{F}_{L_1}, \quad Y_2 = g_2(X); g_2 \in \mathcal{G}_{L_2}; \mathcal{G}_{L_2} \subset \mathcal{F}_{L_2}, \tag{95}$$

*where $\mathcal{G}_{L_1}$ and $\mathcal{G}_{L_2}$ are subsets corresponding to ResNets with 1-Lipschitz activations of $L_1$ and $L_2$ layers, respectively. Further assume that the kernels $\{W_i^{(l)}\}_{i=1}^{D}; W_i \in \mathbb{R}^{K \times K}$, for the convolutional layers $l = \{1, ..., \max\{L_1, L_2\}\}$ in $\mathcal{G}_{L_1}$ and $\mathcal{G}_{L_2}$, are bounded such that:*

$$\sup_{l=1}^{\max\{L_1, L_2\}} \sup_{i=1}^{D} \sum_{j=1}^{K} \sum_{k=1}^{K} \left|\left[W_i^{(l)}\right]_{j,k}\right| \leq \gamma, \tag{96}$$

*where $\gamma \geq 0$. Let $\mathcal{D}_X = \{\mathbf{x}_1\}_{i=1}^{N} \sim X$ be a set of samples from $X$. Then, $\forall \delta \in (0, 1)$, with probability at least $1 - \delta$, we have:*

$$C_{\mathcal{S}}(\mathcal{D}; Y_1, Y_2) \leq 2\left[\mu(1+\gamma)^{L_1} + (1+\gamma)^{L_2}\right]\mathrm{Rad}(\mathcal{D}_X) + B\sqrt{2/N \log 2/\delta}. \tag{97}$$

*Proof.*

$$C_S(\mathcal{D}; Y_1, Y_2) \leq 2\left[\mu\mathrm{Rad}(\mathcal{D}_1) + \mathrm{Rad}(\mathcal{D}_2)\right] + B\sqrt{2/N \log 2/\delta} \tag{98}$$

$$= 2\left[\mu\mathrm{Rad}_{\mathcal{D}_X}(\mathcal{G}_{L_1}) + \mathrm{Rad}_{\mathcal{D}_X}(\mathcal{G}_{L_2})\right] + B\sqrt{2/N \log 2/\delta} \tag{99}$$

$$\leq 2\left[\mu\mathrm{Lip}(\mathcal{G}_{L_1}) + \mathrm{Lip}(\mathcal{G}_{L_2})\right]\mathrm{Rad}(\mathcal{D}_X) + B\sqrt{2/N \log 2/\delta} \tag{100}$$

$$\leq 2\left[\mu(1+\gamma)^{L_1} + (1+\gamma)^{L_2}\right]\mathrm{Rad}(\mathcal{D}_X) + B\sqrt{2/N \log 2/\delta}. \tag{101}$$

Eq. 98 uses Theorem 3. Eq. 100 uses the Kakade & Tewari Lemma previously introduced. Eq. 101 uses Lemma 2 in Behrmann et al. (2019), where for any functions $f$ and $g$, such that $f(x) = x + g(x)$ with $\mathrm{Lip}(g) = \gamma$ Then, it holds that $\mathrm{Lip}(f) \leq 1 + \gamma$. The Lipschitz constant for a convolutional layer with 1-Lipschitz activations (e.g. ReLU) is defined in Truong (2022) and shown in Eq. 96. We also use $\mathrm{Lip}(g \circ f) = \mathrm{Lip}(g)\mathrm{Lip}(f)$, for any Lipschitz functions $g$ and $f$. □

## C.1 ANALYSIS

Theorem 4 shows that, when $\gamma \neq 0$, the compatibility between representations $Y_1$ and $Y_2$ decreases with the depth of the ResNets producing them. For representations that share a neural network, such that the representations $Y_1$ and $Y_2$ are splits of its output, $X$ corresponds to the output of the next to last layer and hence the neural network producing $Y_1$ and $Y_2$ would only have one layer, increasing their compatibility.

In our problem space, it is desirable for the common representation to be compatible with the private representations. The Separable approach has a dedicated ResNet for each representation, which produces less compatible representations compared to our approach, in which each private representation shares most of a ResNet with a copy of the common representation.

In the Combined approach, all representations share most of a ResNet, which increases compatibility among them, but constraints the private channels to produce similar information since they share many of the intermediate representations. A Combined approach with higher dimensionality could compensate for this limitation, but it would be more inefficient than the proposed method, since the kernels would have to be larger in order to accommodate for the larger dimensionality, and parts of the input for each layer might be irrelevant to some of the target representations.

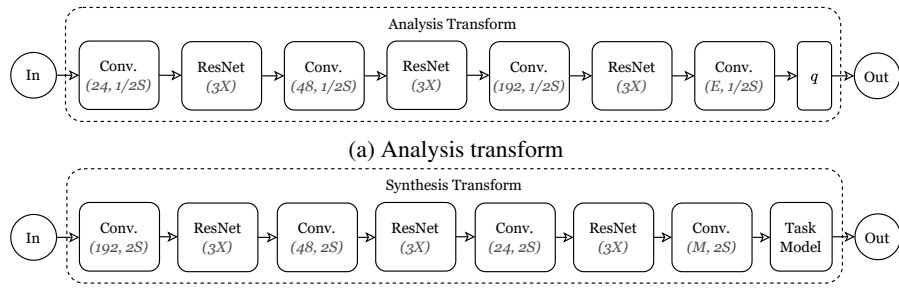

(a) Analysis transform

(b) Synthesis transform

Figure 6: Architecture diagrams of the analysis and synthesis transforms. The number of channels in the input is denoted as $M$. The values assigned to $S$ corresponds to a multiplying change in the spatial dimensions. The values assigned to $X$ represents the number of blocks. The values 24, 48, and 192 correspond to the number of channels in the output representation of the module.

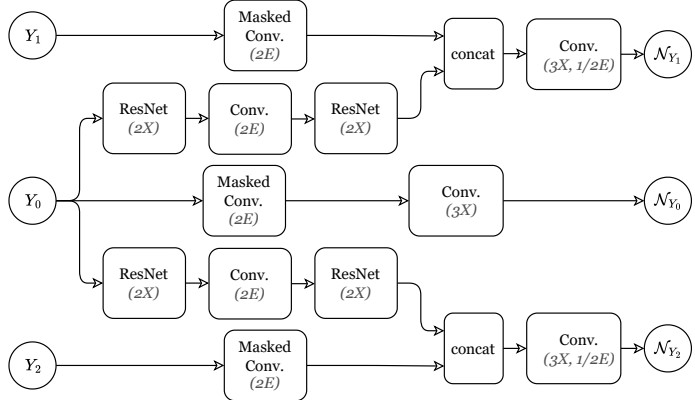

Figure 7: Architecture overview of the three entropy models. $\mathcal{N}_{Y_{\{0,1,2\}}}$ represents the multivariate normal parameters with diagonal covariances assigned to $Y_{\{0,1,2\}}$. They define $\tilde{P}_{Y_{\{0,1,2\}}}$. The values assigned to $X$ represents the number of blocks. The values assigned to $E$ correspond to a multiplying change in the number of channels. For example, "Conv. *(3X)*" corresponds to 3 convolutional layers, and "Conv. *(2E)*" corresponds to a convolutional layer that doubles the number of channels.

## D    ADDITIONAL DETAILS AND RESULTS

### D.1    ARCHITECTURE DETAILS

Figure 6 shows architecture diagrams of the analysis and synthesis transforms. Figure 7 shows an architecture diagram of the three entropy models used in the proposed method. All ResNets use ELU activation functions. The number of channels in the intermediate representation in the ResNet blocks have the same number of channels as the input (and output).

Figure 8 presents an overview of the codec architectures used as benchmarks. The entropy model architecture used for the three-channel codecs is the same one described for the proposed architecture. The two-channel and one-channel codecs have the same entropy model architecture as the one used for the $Y_0$ representation in the proposed architecture.

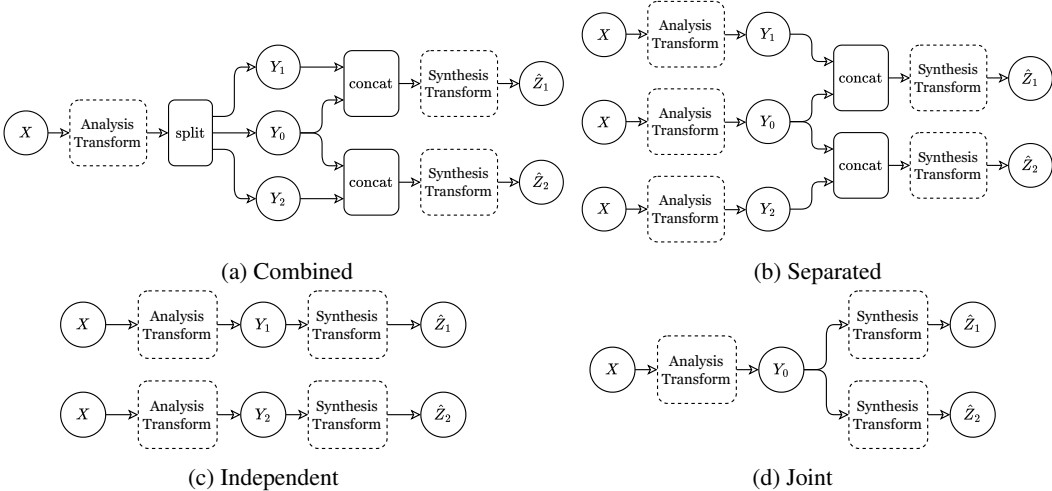

Figure 8: Architecture diagrams of the codecs used as benchmarks. The Combined and Separated approaches have three channels each, and are considered different versions of a Gray-Wyner Network. The Independent codec has two channels, and the Joint codec only has one. For simplicity, the entropy models that code $Y_{\{0,1,2\}}$ are omitted.

Table 1: Hyper-parameter and training settings

| Parameter | Synthetic | MNIST | Cityscapes | COCO 2017 |
|---|---|---|---|---|
| Precision | float32 | float32 | float32 | float32 |
| Input representation channels ($M$) | 2 | 3 | 3 | 3 |
| Target representation channels ($E$) | 512 | 6 | 192 | 192/288 |
| Auxiliary loss weight ($\gamma$) | 1 | 1 | 1 | 1 |
| Reconstruction loss weight | 0 | 0 | 1 | 1 |
| Random horizontal flip | No | No | Yes | Yes |
| Color jittering | No | No | Yes | No |
| Batch size | 100 | 100 | 4 | 4 |
| Accumulated gradient batches | 1 | 1 | 3 | 1 |
| Optimizer | Adam | Adam | Adam | Adam |
| Optimizer parameters | 0.9, 0.999 | 0.9, 0.999 | 0.9, 0.999 | 0.9, 0.999 |
| Learning rate | 0.0001 | 0.0001 | 0.0001 | 0.0001 |
| Weight decay | 0 | 0 | 0 | 0 |
| Gradient norm clipping | 1 | 1 | 1 | 1 |
| Patience | 10 | 25 | 50 | 5 |
| Maximum number of epochs | 100 | 200 | 1,000 | 100 |

### D.2 TRAINING DETAILS

Since the task losses (distortion functions) used are on a similar scale, we only consider rate-distortion points where $\lambda_1 = \lambda_2$. Thus, we scale the optimization objective $\mathcal{L}$ by $\eta = 1/\lambda_{\{1,2\}}$ to obtain a single hyper-parameter controlling the rate-distortion tradeoff.

Table 1 shows the training details used in all experiments. Additional considerations for reproducibility are addressed in other related sections.

### D.3 TRANSMIT-RECEIVE TRADEOFF AND ABLATION STUDY

Elements of a random variable $X$ with sample space $\{-4,...,4\}^{2\times64\times64}$ are identically distributed according to a probability mass function (PMF) created from the normal distribution $\mathcal{N}(0,4)$, which

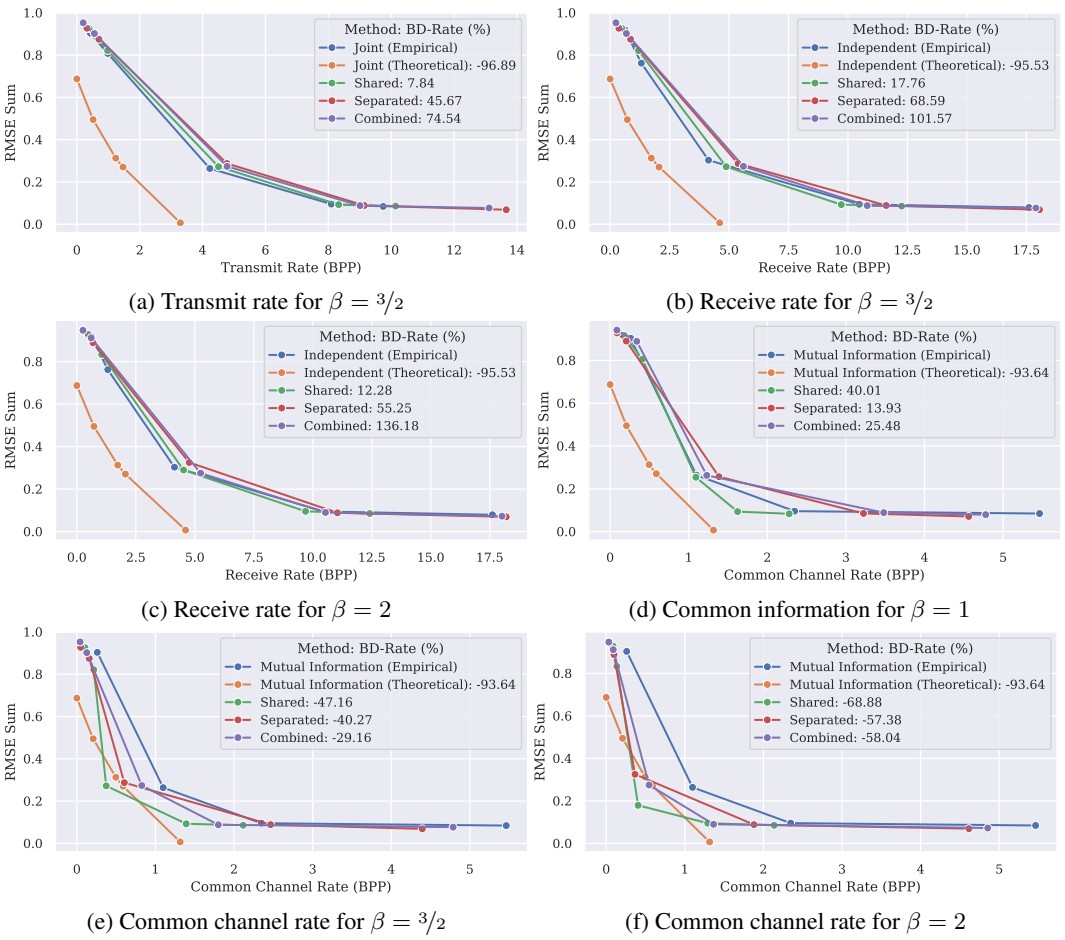

Figure 9: Additional rate-distortion curves for different methods, optimized for transmit, receive, and mixed rates. The common information axes correspond to the bitrates of the common channel, the theoretical measurements of mutual information, and its empirical estimates. BD-rates (Bjontegaard, 2001) are computed with respect to the method with no score. Bits-per-pixel (BPP) is the bitrate scaled by $1/64 \times 64$.

Table 2: Rate-distortion performance of baselines on synthetic dataset

| Codec | $\beta$ | $\eta$ | $R_1$ | $R_2$ | $R_0$ | $R_t$ | $R_r$ | $D_1$ | $D_2$ |
|---|---|---|---|---|---|---|---|---|---|
| Independent | N/A | 0.001 | 8.699 | 8.913 | 0 | 17.612 | 17.612 | 0.040 | 0.039 |
| Independent | N/A | 0.01 | 5.047 | 5.424 | 0 | 10.471 | 10.471 | 0.046 | 0.048 |
| Independent | N/A | 0.1 | 2.077 | 2.059 | 0 | 4.136 | 4.136 | 0.152 | 0.151 |
| Independent | N/A | 0.5 | 0.656 | 0.655 | 0 | 1.311 | 1.311 | 0.381 | 0.381 |
| Independent | N/A | 1.0 | 0.287 | 0.334 | 0 | 0.621 | 0.621 | 0.459 | 0.460 |
| Joint | N/A | 0.001 | 0 | 0 | 9.745 | 9.745 | 19.49 | 0.042 | 0.043 |
| Joint | N/A | 0.01 | 0 | 0 | 8.096 | 8.096 | 16.192 | 0.048 | 0.048 |
| Joint | N/A | 0.1 | 0 | 0 | 4.230 | 4.230 | 8.46 | 0.131 | 0.132 |
| Joint | N/A | 0.5 | 0 | 0 | 0.983 | 0.983 | 1.966 | 0.402 | 0.406 |
| Joint | N/A | 1.0 | 0 | 0 | 0.429 | 0.429 | 0.858 | 0.452 | 0.451 |

is first clipped to $[-4, 4]$, and then quantized to 9 symbols using the integers in the range. Contiguous element pairs were either selected to be copies, with probability $4/5$, or independent, with probability $1/5$. The random variable positions were then shuffled and the resulting dependency map was stored and used to generate all samples from $X$.

Table 3: Rate-distortion performance of proposed methods on synthetic dataset for $\beta = 1$

| Codec | $\beta$ | $\eta$ | $R_1$ | $R_2$ | $R_0$ | $R_t$ | $R_r$ | $D_1$ | $D_2$ |
|---|---|---|---|---|---|---|---|---|---|
| Shared | 1 | 0.001 | 3.747 | 4.419 | 2.278 | 10.444 | 12.722 | 0.042 | 0.041 |
| Shared | 1 | 0.01 | 3.110 | 3.639 | 1.622 | 8.371 | 9.993 | 0.046 | 0.048 |
| Shared | 1 | 0.1 | 1.504 | 2.056 | 1.091 | 4.651 | 5.742 | 0.128 | 0.126 |
| Shared | 1 | 0.5 | 0.247 | 0.509 | 0.410 | 1.166 | 1.576 | 0.403 | 0.403 |
| Shared | 1 | 1.0 | 0.050 | 0.230 | 0.180 | 0.460 | 0.640 | 0.461 | 0.456 |
| Separated | 1 | 0.001 | 4.287 | 4.604 | 4.563 | 13.454 | 18.017 | 0.036 | 0.035 |
| Separated | 1 | 0.01 | 2.729 | 3.201 | 3.224 | 9.154 | 12.378 | 0.042 | 0.042 |
| Separated | 1 | 0.1 | 1.781 | 1.792 | 1.384 | 4.957 | 6.341 | 0.129 | 0.128 |
| Separated | 1 | 0.5 | 0.212 | 0.233 | 0.205 | 0.650 | 0.855 | 0.445 | 0.445 |
| Separated | 1 | 1.0 | 0.106 | 0.134 | 0.088 | 0.328 | 0.416 | 0.467 | 0.462 |
| Combined | 1 | 0.001 | 4.130 | 4.029 | 4.778 | 12.937 | 17.715 | 0.039 | 0.040 |
| Combined | 1 | 0.01 | 2.807 | 2.749 | 3.481 | 9.037 | 12.518 | 0.044 | 0.045 |
| Combined | 1 | 0.1 | 1.880 | 1.794 | 1.230 | 4.904 | 6.134 | 0.131 | 0.131 |
| Combined | 1 | 0.5 | 0.233 | 0.201 | 0.339 | 0.773 | 1.112 | 0.443 | 0.447 |
| Combined | 1 | 1.0 | 0.095 | 0.097 | 0.086 | 0.278 | 0.364 | 0.471 | 0.470 |

Table 4: Rate-distortion performance of proposed methods on synthetic dataset for $\beta = 3/2$

| Codec | $\beta$ | $\eta$ | $R_1$ | $R_2$ | $R_0$ | $R_t$ | $R_r$ | $D_1$ | $D_2$ |
|---|---|---|---|---|---|---|---|---|---|
| Shared | 3/2 | 0.001 | 3.662 | 4.363 | 2.116 | 10.141 | 12.257 | 0.043 | 0.043 |
| Shared | 3/2 | 0.01 | 3.238 | 3.698 | 1.392 | 8.328 | 9.720 | 0.047 | 0.046 |
| Shared | 3/2 | 0.1 | 2.007 | 2.125 | 0.373 | 4.505 | 4.878 | 0.137 | 0.136 |
| Shared | 3/2 | 0.5 | 0.295 | 0.468 | 0.215 | 0.978 | 1.193 | 0.413 | 0.409 |
| Shared | 3/2 | 1.0 | 0.066 | 0.222 | 0.104 | 0.392 | 0.496 | 0.471 | 0.455 |
| Separated | 3/2 | 0.001 | 4.518 | 4.746 | 4.395 | 13.659 | 18.054 | 0.034 | 0.035 |
| Separated | 3/2 | 0.01 | 3.382 | 3.295 | 2.465 | 9.142 | 11.607 | 0.044 | 0.045 |
| Separated | 3/2 | 0.1 | 2.052 | 2.118 | 0.604 | 4.774 | 5.378 | 0.143 | 0.145 |
| Separated | 3/2 | 0.5 | 0.277 | 0.269 | 0.157 | 0.703 | 0.860 | 0.435 | 0.441 |
| Separated | 3/2 | 1.0 | 0.140 | 0.132 | 0.049 | 0.321 | 0.370 | 0.463 | 0.464 |
| Combined | 3/2 | 0.001 | 4.254 | 4.071 | 4.788 | 13.113 | 17.901 | 0.039 | 0.038 |
| Combined | 3/2 | 0.01 | 3.618 | 3.591 | 1.799 | 9.009 | 10.809 | 0.044 | 0.044 |
| Combined | 3/2 | 0.1 | 1.996 | 1.957 | 0.826 | 4.779 | 5.605 | 0.137 | 0.137 |
| Combined | 3/2 | 0.5 | 0.215 | 0.216 | 0.126 | 0.557 | 0.683 | 0.451 | 0.451 |
| Combined | 3/2 | 1.0 | 0.083 | 0.078 | 0.041 | 0.202 | 0.243 | 0.476 | 0.477 |

Two bijective linear functions of determinant 1 are arbitrarily generated for $X_1$ and $X_2$, respectively. The output of these transformations are the corresponding targets $Z_1$ and $Z_2$ for two linear regression tasks. We dynamically generate a dataset of infinite samples from $P_{X,Z_1,Z_2}$ on the fly.

The mutual information $I(X_1; X_2)$ increases with the number of elements in $X_1$ that are duplicated in $X_2$. Using the Blahut–Arimoto algorithm (Cover & Thomas, 2006), we generate theoretical rate-distortion curves for $X$. The resulting PMF of the input reconstruction $\hat{X}$ allows the computation of $H(\hat{Z}_1, \hat{Z}_2)$, $H(\hat{Z}_1) + H(\hat{Z}_2)$, and $I(\hat{Z}_1; \hat{Z}_2)$, for the corresponding distortion values $D_1$ and $D_2$.

To obtain empirical estimates of mutual information, we use the rate of the single channel in the Joint method as an empirical measurement $\tilde{H}(\hat{Z}_1, \hat{Z}_2)$. We use the rates of the Independent method as empirical values $\tilde{H}(\hat{Z}_1) + \tilde{H}(\hat{Z}_2)$. To compute an empirical estimation of the mutual information between tasks, we interpolate and extrapolate the points within the rate-distortion curves of these methods, covering all distortion points. Then, we compute $\hat{I}(\hat{Z}_1; \hat{Z}_2) = \tilde{H}(\hat{Z}_1) + \tilde{H}(\hat{Z}_2) - \tilde{H}(\hat{Z}_1, \hat{Z}_2)$ between points with the same distortion value.

To improve task performance, all scaling convolutional layers are replaced with *pixel shuffle* operations (Shi et al., 2016), removing dimensionality bottlenecks in the architecture. Each pixel shuffle

Table 5: Rate-distortion performance of proposed methods on synthetic dataset for $\beta = 2$

| Codec | $\beta$ | $\eta$ | $R_1$ | $R_2$ | $R_0$ | $R_t$ | $R_r$ | $D_1$ | $D_2$ |
|-------|---------|--------|-------|-------|-------|-------|-------|-------|-------|
| Shared | 2 | 0.001 | 3.656 | 4.48 | 2.137 | 10.273 | 12.41 | 0.044 | 0.041 |
| Shared | 2 | 0.01 | 3.332 | 3.777 | 1.291 | 8.4 | 9.691 | 0.049 | 0.046 |
| Shared | 2 | 0.1 | 2.116 | 2.139 | 0.133 | 4.388 | 4.521 | 0.143 | 0.147 |
| Shared | 2 | 0.5 | 0.305 | 0.475 | 0.136 | 0.916 | 1.052 | 0.424 | 0.41 |
| Shared | 2 | 1.0 | 0.066 | 0.239 | 0.087 | 0.392 | 0.479 | 0.474 | 0.455 |
| Separated | 2 | 0.001 | 4.494 | 4.474 | 4.615 | 13.583 | 18.198 | 0.034 | 0.035 |
| Separated | 2 | 0.01 | 3.748 | 3.535 | 1.881 | 9.164 | 11.045 | 0.044 | 0.045 |
| Separated | 2 | 0.1 | 2.016 | 2.015 | 0.366 | 4.397 | 4.763 | 0.16 | 0.166 |
| Separated | 2 | 0.5 | 0.168 | 0.319 | 0.099 | 0.586 | 0.685 | 0.46 | 0.428 |
| Separated | 2 | 1.0 | 0.075 | 0.115 | 0.029 | 0.219 | 0.248 | 0.478 | 0.47 |
| Combined | 2 | 0.001 | 4.215 | 4.095 | 4.854 | 13.164 | 18.018 | 0.036 | 0.037 |
| Combined | 2 | 0.01 | 3.863 | 3.94 | 1.366 | 9.169 | 10.535 | 0.045 | 0.045 |
| Combined | 2 | 0.1 | 2.11 | 2.042 | 0.545 | 4.697 | 5.242 | 0.137 | 0.138 |
| Combined | 2 | 0.5 | 0.221 | 0.207 | 0.09 | 0.518 | 0.608 | 0.451 | 0.46 |
| Combined | 2 | 1.0 | 0.099 | 0.094 | 0.033 | 0.226 | 0.259 | 0.473 | 0.474 |

Table 6: Rate-distortion performance of the proposed method on colored MNIST

| Codec | $\eta$ | $R_1$ | $R_2$ | $R_0$ | $R_t$ | $R_r$ | $1/D_1$ | $1/D_2$ |
|-------|--------|-------|-------|-------|-------|-------|---------|---------|
| Dependent | 1 | 2.908 | 5.779 | 15.84 | 24.526 | 40.366 | 0.999 | 0.999 |
| Dependent | 10 | 1.460 | 4.071 | 10.641 | 16.172 | 26.812 | 0.998 | 0.999 |
| Dependent | 25 | 1.571 | 2.637 | 10.524 | 14.731 | 25.255 | 0.996 | 0.998 |
| Dependent | 50 | 1.267 | 2.500 | 9.921 | 13.688 | 23.609 | 0.995 | 0.996 |
| Dependent | 75 | 2.186 | 0.826 | 10.366 | 13.378 | 23.743 | 0.992 | 0.989 |
| Dependent | 100 | 1.067 | 2.318 | 8.991 | 12.376 | 21.368 | 0.975 | 0.986 |
| Independent | 1 | 22.269 | 6.864 | 0.071 | 29.203 | 29.273 | 0.989 | 0.999 |
| Independent | 10 | 15.974 | 7.030 | 0.056 | 23.060 | 23.116 | 0.983 | 0.999 |
| Independent | 25 | 13.568 | 5.656 | 0.017 | 19.241 | 19.257 | 0.975 | 0.996 |
| Independent | 50 | 12.073 | 5.776 | 0.019 | 17.868 | 17.887 | 0.970 | 0.969 |
| Mixture | 1 | 19.23 | 8.425 | 5.149 | 32.804 | 37.952 | 0.993 | 1 |
| Mixture | 10 | 9.322 | 5.349 | 10.966 | 25.637 | 36.603 | 0.992 | 1 |
| Mixture | 25 | 9.965 | 4.684 | 10.025 | 24.674 | 34.699 | 0.989 | 0.999 |
| Mixture | 100 | 9.547 | 3.660 | 9.629 | 22.835 | 32.464 | 0.975 | 0.997 |
| Mixture (Independent) | 1 | 27.262 | 15.658 | 0 | 42.919 | 42.919 | 0.991 | 0.999 |
| Mixture (Independent) | 10 | 16.182 | 18.592 | 0 | 34.773 | 34.773 | 0.988 | 0.999 |
| Mixture (Independent) | 25 | 16.031 | 15.464 | 0 | 31.495 | 31.495 | 0.979 | 0.999 |
| Mixture (Independent) | 50 | 14.913 | 13.892 | 0 | 28.805 | 28.805 | 0.977 | 0.998 |
| Mixture (Independent) | 75 | 13.804 | 14.685 | 0 | 28.489 | 28.489 | 0.967 | 0.994 |

or unshuffle operation has a factor of 2, which implies that the analysis transform progressively increases the number of channels from 2 to 8, 32, 128, and finally $E = 512$. The synthesis transform decreases the number of channel in reverse order. We use 1,000 steps per training epoch and 100 per validation epoch.

Figure 9 shows additional results. The transmit and receive rate plots show that the Shared (proposed) method outperforms the Separated and Combined baselines. The common channel rate plots show in detail how $\beta$ controls the amount of information in the common channel. Tables 2, 3, 4, 5 show the values used in these plots. All rate values are bitrates scaled by $1/64 \times 64$.

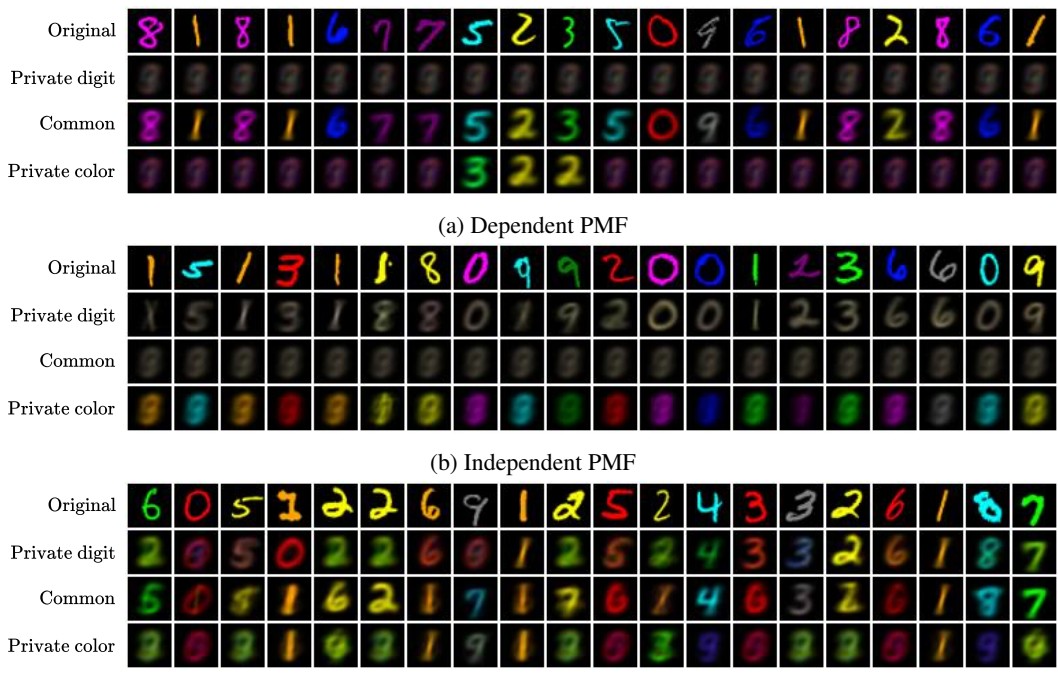

(a) Dependent PMF

(b) Independent PMF

(c) Mixture PMF

Figure 10: MNIST image reconstructions from individual channels $Y_{\{0,1,2\}}$ produced by codecs trained on one of the 3 different PMFs discussed. The Dependent PMF places most information on the common channel. The Independent PMF places the digit and color information on the corresponding private channels. In this codec, the common channel does not seem to carry any information. The Mixture PMF shows a combination of information across channels. The correct answer is usually found in a combination of channels.

### D.4    EDGE-CASE EXPLORATION WITH IMAGE CLASSIFICATION

We set the number of dimensions to $E = 6$. Due to the low rate of the channels, the auxiliary loss term in Eq. 15, which encourages the common representations $Y_0^{(1)}$ and $Y_0^{(2)}$ to match, prevents the common channel from carrying any significant amount of information. We overcome this obstacle by setting $\beta = 1/10$, encouraging the use of the common channel, and compensating for the restrictions of the auxiliary loss.

Table 6 shows the values obtained for the different PMFs and methods. Rates are shown as bitrates (using $\log$ functions with base 2). The inverse distortions $1/D_1$ and $1/D_2$ are the top-1 classification accuracies. The Independent method has very low rate on the common channel, as it would be optimal to do so. The Dependent method places some information on the private channels, but it is on average 29.22% of the transmit rate.

To visualize the information available in these channels, we train multiple tasks to reconstruct the input image, each using the representations of a channel as input. The model architecture is the same as the proposed synthesis transform. The kernel sizes of the first and last layers are adapted to dimensional requirements of the input and output, respectively. We train to minimize the RMSE between the predictions and the input image, using the same settings described for MNIST in Section D.2. We generate reconstructed images for each PMF from corresponding codecs that produce the lowest rate.

Figure 10 shows the results from uniform samples of the validation set. For the Dependent and Independent PMFs, it is easily observed that the available information matches the quantitative results. In the results from the Mixture PMF, the color channel does not contain much digit information. The digit channel contains some color information but it is often incorrect. Whenever a private channel is incorrect, the common channel seems to contain the correct answer.

Table 7: Rate-distortion performance of the proposed method on Cityscapes

| Codec | $\eta$ | $R_1$ | $R_2$ | $R_0$ | $R_t$ | $R_r$ | $1/D_1$ | $D_2$ |
|---|---|---|---|---|---|---|---|---|
| Joint | 0.001 | 0 | 0 | 2.444 | 2.444 | 4.888 | 0.670 | 5.493 |
| Joint | 0.01 | 0 | 0 | 2.069 | 2.069 | 4.138 | 0.669 | 5.489 |
| Joint | 0.05 | 0 | 0 | 1.611 | 1.611 | 3.222 | 0.670 | 5.497 |
| Joint | 0.1 | 0 | 0 | 1.527 | 1.527 | 3.054 | 0.669 | 5.496 |
| Joint | 1 | 0 | 0 | 1.074 | 1.074 | 2.148 | 0.663 | 5.497 |
| Independent | 0.001 | 2.113 | 2.296 | 0 | 4.409 | 4.409 | 0.666 | 5.477 |
| Independent | 0.01 | 1.808 | 1.892 | 0 | 3.700 | 3.700 | 0.668 | 5.480 |
| Independent | 0.05 | 1.227 | 1.337 | 0 | 2.564 | 2.564 | 0.665 | 5.493 |
| Independent | 0.1 | 1.104 | 1.191 | 0 | 2.295 | 2.295 | 0.663 | 5.493 |
| Independent | 1 | 0.788 | 0.846 | 0 | 1.634 | 1.634 | 0.660 | 5.513 |
| Proposed | 0.001 | 1.384 | 1.482 | 0.330 | 3.196 | 3.526 | 0.666 | 5.482 |
| Proposed | 0.01 | 1.329 | 1.415 | 0.243 | 2.987 | 3.230 | 0.669 | 5.487 |
| Proposed | 0.05 | 1.217 | 1.285 | 0.329 | 2.831 | 3.160 | 0.669 | 5.488 |
| Proposed | 0.1 | 1.110 | 1.151 | 0.414 | 2.675 | 3.089 | 0.669 | 5.487 |
| Proposed | 1.0 | 0.568 | 0.568 | 0.409 | 1.545 | 1.954 | 0.666 | 5.488 |
| Proposed | 5.0 | 0.315 | 0.311 | 0.319 | 0.945 | 1.264 | 0.662 | 5.496 |

Table 8: Rate-distortion performance of the proposed method on COCO 2017

| Codec | $\eta$ | $R_1$ | $R_2$ | $R_0$ | $R_t$ | $R_r$ | $1/D_1$ | $1/D_2$ |
|---|---|---|---|---|---|---|---|---|
| Joint | 0.001 | 0 | 0 | 3.513 | 3.513 | 7.026 | 0.453 | 0.646 |
| Joint | 0.05 | 0 | 0 | 3.198 | 3.198 | 6.396 | 0.454 | 0.643 |
| Joint | 0.1 | 0 | 0 | 2.997 | 2.997 | 5.994 | 0.453 | 0.642 |
| Joint | 1 | 0 | 0 | 1.980 | 1.980 | 3.960 | 0.449 | 0.638 |
| Independent | 0.001 | 3.676 | 3.543 | 0 | 7.219 | 7.219 | 0.453 | 0.645 |
| Independent | 0.01 | 3.498 | 3.264 | 0 | 6.762 | 6.762 | 0.453 | 0.644 |
| Independent | 0.05 | 3.113 | 2.973 | 0 | 6.086 | 6.086 | 0.454 | 0.643 |
| Independent | 0.1 | 2.855 | 2.768 | 0 | 5.623 | 5.623 | 0.453 | 0.643 |
| Independent | 1 | 1.615 | 1.605 | 0 | 3.220 | 3.220 | 0.448 | 0.635 |
| Proposed | 0.001 | 2.725 | 2.681 | 1.482 | 6.888 | 8.370 | 0.454 | 0.646 |
| Proposed | 0.5 | 1.532 | 1.559 | 1.002 | 4.093 | 5.095 | 0.455 | 0.646 |
| Proposed | 1 | 1.230 | 1.250 | 0.842 | 3.322 | 4.164 | 0.452 | 0.643 |
| Proposed | 2.5 | 0.822 | 0.845 | 0.659 | 2.326 | 2.985 | 0.449 | 0.637 |

## D.5 RATE-DISTORTION PERFORMANCE IN COMPUTER VISION TASKS

For semantic segemenation on Cityscapes, we use pretrained weights provided by Chen et al. (2018a). For the object detection and keypoint detection tasks on COCO 2017, we use pretrained weights provided by TorchVision maintainers and contributors (2016). The depth estimation model is trained from scratch without rate constraints and then used as a frozen pretrained model in the proposed method. Since the size of the Cityscape images is quite large, for tractability, we learn a downsampler of the input images and an upsampler of the output predictions of the task model, for the semantic segmentation and depth estimation task models. The task distortions are still computed against the original Cityscapes targets. The factor of the samplers is 2.

We set $E = 192$ for all models except those trained on COCO 2017 with the proposed method, which required $E = 288$. Since the task models are frozen, a reconstruction loss between the input and the output of the synthesis transforms, before the task models, improves rate-distortion performance. The weight of this reconstruction loss is listed in Table 1.

We note that the rates produced by our methods are relatively large. This is attributed to the synthesis transform being connected to the first layer of the task models and them remaining frozen, effec-

tively encouraging the reconstruction of the input images. Since the images are highly compressed JPEG images, they contain many artifacts that might need to be reconstructed, requiring additional rate. Interestingly, the input reconstruction loss seems to improve rate-distortion performance, even though intuitively one might think it would not.

Tables 7 and 8 show the values obtained by our proposed method against the Independent and Joint baselines. All results were collected from validation sets. The rates are reported in bits-per-pixel (BPP). In the Cityscapes experiments, $1/D_1$ and $D_2$ correspond to the mIoU and RMSE metrics, respectively. In the COCO 2017 experiments, $1/D_1$ and $1/D_2$ correspond to the object detection mAP and keypointing mAP metrics, respectively.

All experiments are trained with $\beta = 1$, which, as previously discussed, due to the constraints of the auxiliary loss, does not necessarily optimize for the transmit rate. In fact, since the results show that the receive rate of the proposed method is better than the Independent method, the common channel must not contain all empirical mutual information possible.

