# OpenReview forum: "Lossy Common Information in a Learnable Gray-Wyner Network"
_ICLR.cc/2026/Conference — ICLR 2026 Poster_

### Official Review · Reviewer_wYFu · 2025-10-27

**Soundness:** 3
**Presentation:** 1
**Contribution:** 3
**Rating:** 6
**Confidence:** 3

**Summary:**

This paper provides a theoretically grounded and empirically validated framework that connects Gray–Wyner information theory with modern learnable architectures. While the architectural novelty is moderate and writing can be improved, the theoretical novelty, empirical consistency, and conceptual clarity make it a strong contribution.

**Strengths:**

1.	The theoretical analysis appears rigorous and well-presented. I did not identify mistakes or inconsistencies in the proofs of Theorem 1 and Theorem 2.
2.	Addressing the challenge of distilling shared information in multi-task learning is both important and underexplored. The proposed learnable Gray-Wyner Network provides a natural and conceptually clear solution, supported by solid theoretical justification and empirical evidence.
3.	I appreciate the authors’ effort to draw inspiration from classical information-theoretic principles (i.e., the Gray-Wyner framework) and adapt them to modern machine learning contexts.
4.	The empirical results are consistent with the theoretical analysis and convincingly demonstrate the effectiveness of the proposed method.

**Weaknesses:**

1.	My main concern lies in the presentation of the paper. The notations are dense and difficult to follow, particularly in Sections 2.1 and 3.1. The motivation behind the proposed method is somewhat unclear. Reorganizing the theoretical sections with more intuitive explanations and a clearer logical flow would significantly improve readability.
2.	Although the method is designed to “disentangle” shared information, the paper lacks visualization or analysis to demonstrate that the learned representations indeed capture task-common semantics.
3.	It would be beneficial to include error bars in the experimental results to reflect the variance and enhance the reliability of the reported performance.

**Questions:**

See Weaknesses.

---

> ### Author Response · Authors · 2025-11-18
> **Opening Message**
>
> We appreciate the reviewers for taking their time to analyze the paper and give thoughtful, actionable, feedback. We understand that the nature of our work can be considered niche to the ICLR community and appreciate that the reviewers took extra effort to get acquainted with it. We believe that information theory is the cornerstone of representation learning and, as such, ICLR is the best venue for our work.
>
> Please see the additional messages for our responses to each of the reviewers' comments. We apologize for the several notifications they might receive. Since some of the suggestions/weaknesses are not numbered, we decided to include them in each comment. We have submitted a PDF as supplementary material highlighting the content differences between the submitted version and the rebuttal version. The new content is highlighted in blue, whereas the erased content is highlighted in red.
>
> We hope to have addressed most concerns satisfactorily and we encourage reviewers to follow up on the discussion.

---

> > ### Author Response · Authors · 2025-11-18
> >
> > ### My main concern lies in the presentation of the paper. The notations are dense and difficult to follow, particularly in Sections 2.1 and 3.1. The motivation behind the proposed method is somewhat unclear. Reorganizing the theoretical sections with more intuitive explanations and a clearer logical flow would significantly improve readability.
> >
> > Thank you, we agree that the initial submission was a bit dense. Taking advantage of the additional page available for the rebuttal, we focused on improving the clarity of the paper. As per the reviewer's suggestions, we added intuitive explanations in Sections 2.1 (Preliminaries) and 3.1 (Bounds for lossy common information). There is now a more natural flow from one contribution to the other, where the previous contribution clearly motivates the next one. Among several changes, we:
> >
> > - Justified the information provided as preliminaries.
> > - Provided intuition for the rate-distortion function.
> > - Added lower bounds to the definitions of transmit and receive rates and explained early-on their connection to the common information measures discussed as well as the receive-rate tradeoff.
> > - Explained interaction information, a measure used for the bounds in Theorem 1.
> > - Motivated the importance of Theorem 1 by highlighting the gap between the bounds.
> > - Explained how this gap motivates the exploration of the transmit-receive tradeoff.
> > - Provided more intuition for the transmit-receive tradeoff.
> > - Added a brief introduction to Section 3.3 (A learnable Gray-Wyner Network).
> > - Spelled out the proposed design decisions that are driven by the theoretically-grounded objective function.
> >
> > To motivate the contributions a bit further, we added the following to Section 5 (Summary and conclusion):
> >
> > > "Isolating and coding the common information between dependent tasks allows for the efficient distributed inference of machine tasks. Generating representations that explore the tradeoff between the transmit and receive rates in the Gray-Wyner Network has additional practical implications in storage and selective retrieval, and dispersive information routing (Viswanatha et al., 2011). Knowing the information requirements of learned representations can assist in planning for the resources allocated to a neural network, its dimensionality, and quantization levels."

---

> > ### Author Response · Authors · 2025-11-18
> >
> > ### Although the method is designed to “disentangle” shared information, the paper lacks visualization or analysis to demonstrate that the learned representations indeed capture task-common semantics.
> >
> > We refer the reviewer to Appendix D.3. (Edge-case exploration with image classification), in which, for the Dependent, Independent, and Mixture PMFs of the colored MNIST dataset, we show the results of reconstructing the original input from the private and common channels, individually. The results visually show the separation of information for the Independent PMF, and the consolidation of most information in the common channel under the Dependent PMF.
> >
> > We did not apply this technique to the non-trivial computer visions tasks of Section 4.3 (Rate-distortion performance in computer vision tasks) since we do not know what to expect from such reconstructions and they might be too marginalized to appreciate any differences.
> >
> > Unfortunately, we were not able to include these results in the main sections since they take too much space. We have used most of the additional page available for the rebuttal to improve clarity and refine other sections, which we believe to be more crucial.

---

> ### Author Response · Authors · 2025-11-18
>
> ### It would be beneficial to include error bars in the experimental results to reflect the variance and enhance the reliability of the reported performance.
>
> We acknowledge that providing error bars would increase the statistical significance of the results. Unfortunately, computing these would be prohibitively costly for us. Each anchor point on the rate-distortion curves presented correspond to one fully-trained model. Moreover, since there is variation in both rate and distortion (task accuracy), error bars would have to be reported along both axes, which would complicate interpretation. For this reason, error bars are usually not computed or reported in the compression literature. In this work, we are showing the results from approximately 98 different models. Because we wanted to demonstrate the reliability of our methods and insights, we performed experiments for a variety of tasks and datasets. Our results show consistency in findings that agree with our theory, across all these experiments.
>
> Nevertheless, we have selected a subset of experiments for which we want to compute error bars. However, most likely, the experiment runs will not be finished before the deadline.
>
> Please note that regarding performance variance across runs, all models in the same rate distortion curve run under the same hyper-parameters, except for the rate-distortion tradeoff ($\eta$). Each of these models is trained from a different random parameter initialization. It is often seen in this type of work that if a method produces metrics with significant variance, it is difficult to obtain convex rate-distortion curves. We do not see much of that behaviour with our proposed method.

---

> > ### Comment · Reviewer_wYFu · 2025-11-26
> >
> > Thank you for your detailed and insightful response.
> >
> > Overall, I still view the paper as marginally above the acceptance threshold. I am therefore keeping my rating unchanged, but I have increased my confidence in this assessment.

---

> > > ### Author Response · Authors · 2025-12-03
> > >
> > > Thank you for your response and for increasing your confidence to 4.

---

### Official Review · Reviewer_czUz · 2025-10-27

**Soundness:** 3
**Presentation:** 3
**Contribution:** 3
**Rating:** 6
**Confidence:** 2

**Summary:**

This work proposes a learnable three-channel encoder-decoder based on the Gray-Wyner network from information theory, designed to disentangle shared information and task-specific information in multi-task visual scenarios. The method achieves promising results across multiple benchmarks and settings.

**Strengths:**

1. By analyzing the rate bounds of joint and independent encoding, the method clarifies the feasible conditions for disentangling shared information.
2. A Lagrangian-relaxed optimization objective is designed, allowing the amount of information in the shared channel to be dynamically controlled by the hyperparameter β, supporting a continuous trade-off from transmission-rate optimal (β=1) to reception-rate optimal (β=2).
3. Shared encoding layers extract common information, which is then split into private channels, ensuring representation compatibility while reducing redundancy.
4. The method supports direct integration with pre-trained task models, such as DeepLabV3+ and Faster R-CNN, achieving efficient encoding without the need to fine-tune the task networks.

**Weaknesses:**

1. Theorem 1 provides upper and lower bounds on the amount of shared information, but it is not specified how these bounds can be approached in a practical learning framework. Does this require specific architectural constraints or training mechanisms to ensure tightness of the bounds?
2. The method relies on a Markov assumption that “each source contains no exclusive information that benefits non-corresponding tasks” (Section 2.1). In real-world multi-task learning, cross-task information transfer is often beneficial. This assumption may be overly strict, potentially limiting the applicability of the method.
3. The entropy model for private channels uses the shared channel as context (Section 3.3), which may lead to leakage of shared information into private channels. Are there mechanisms in place to ensure the purity of information separation?
4. The experimental tasks are relatively simple (classification, regression, segmentation). It remains unclear whether the method remains effective for tasks requiring fine-grained spatial reasoning (e.g., instance segmentation) tasks.

**Questions:**

See the Weaknesses.

---

> ### Author Response · Authors · 2025-11-18
> **Opening Message**
>
> We appreciate the reviewers for taking their time to analyze the paper and give thoughtful, actionable, feedback. We understand that the nature of our work can be considered niche to the ICLR community and appreciate that the reviewers took extra effort to get acquainted with it. We believe that information theory is the cornerstone of representation learning and, as such, ICLR is the best venue for our work.
>
> Please see the additional messages for our responses to each of the reviewers' comments. We apologize for the several notifications they might receive. Since some of the suggestions/weaknesses are not numbered, we decided to include them in each comment. We have submitted a PDF as supplementary material highlighting the content differences between the submitted version and the rebuttal version. The new content is highlighted in blue, whereas the erased content is highlighted in red.
>
> We hope to have addressed most concerns satisfactorily and we encourage reviewers to follow up on the discussion.

---

> > ### Author Response · Authors · 2025-11-18
> >
> > ### Theorem 1 provides upper and lower bounds on the amount of shared information, but it is not specified how these bounds can be approached in a practical learning framework. Does this require specific architectural constraints or training mechanisms to ensure tightness of the bounds?
> >
> > Practically speaking, it would be hard for a learnable codec to directly optimize the inner expressions in Theorem 1, as they are defined, since we do not have reliable and tractable ways to measure interaction information in high-dimensional spaces and the rate-distortion constraints are generally unknown. Although some of the Markov conditions in the definitions of common information (the outer terms) can be implemented via architectural constraints, we still face similar issues if we were to directly optimize those definitions instead.
> >
> > Theorem 1 increases our understanding of the problem by showing the connection between the common information measures and the transmit and receive rates. It shows that the transmit-receive tradeoff explores the gap between the bounds. Moreover, it shows the conditions for equality, highlighting that it is harder to achieve than the lossless case, and thus, the gap between the measures can be substantially large. Hence, if we need to optimize for both the transmit and receive rates, we need to explore the gap, since choosing either bound could be highly suboptimal for the other rate.
> >
> > With that in mind, we propose our objective function, which relaxes the rate-distortion constraints, and we also propose a learnable architecture that uses entropy models to learn approximations of the rate, and task models to learn the distortion values. Instead of optimizing for the bounds directly, we optimize for a blend of transmit and receive rates (the tradeoff). We can try to approximate the common information measures if we use values of $\beta$ that closely favour one of the rates (close to 1 for transmit, close to 2 for receive). We cannot fully optimize for either rate though, since it can produce configurations that are nowhere close to the range in Theorem 1.
> >
> > Taking in consideration the suggestions from another reviewer, we have improved on the clarity of the paper. We believe the this more intuitive explanation is now present in the new version, albeit  more formally. Please let us know if you disagree with this sentiment.

---

> > ### Author Response · Authors · 2025-11-18
> >
> > ### The method relies on a Markov assumption that “each source contains no exclusive information that benefits non-corresponding tasks” (Section 2.1). In real-world multi-task learning, cross-task information transfer is often beneficial. This assumption may be overly strict, potentially limiting the applicability of the method.
> >
> > If this Markov assumption does not hold for source random variables $X'_1$ and $X'_2$, it is entirely possible to set $X_1 = X_2 = (X'_1, X'_2)$. If fact, we recommend doing this in our proposed architecture.  We intentionally split $X_1$ and $X_2$ to generalize over this case among others, but we now believe that it might be better to integrate this aspect into the architecture. Thus, in Section 3.3 (A learnable Gray-Wyner Network), we have updated the architecture diagram and the corresponding explanations, and added the following:
> >
> > > "Because each branch of the proposed architecture has access to both sources $X_1$ and $X_2$, all exclusive information from either source is available to assist in performing tasks $Z_1$ or $Z_2$. This effectively removes the requirement for the conditions in 1."
> >
> > We note that the proof for Theorem 1 is simplified in this specialized case but we find value in presenting the results for the generalization. It resembles the lossless counterpart presented in Gray \& Wyner (1974).

---

> > ### Author Response · Authors · 2025-11-18
> >
> > ### The entropy model for private channels uses the shared channel as context (Section 3.3), which may lead to leakage of shared information into private channels. Are there mechanisms in place to ensure the purity of information separation?
> >
> > The private representations might contain information from the common representation but, when the private representations are coded into a bitstream, this redundancy is removed. More specifically, the entropy model is part of the encoder and when it is able to better predict the common information in the private representation, the resulting code will consequentially consume less bits.
> >
> > Removing the common representation context from the private entropy models would encourage the representations to discard unnecessary common information. However, as we have shown, total separation of information is often not possible, and hence, when this cannot be achieved, using an unconditional entropy model would increase the rate of the private channels and make communication inefficient.
> >
> > One approach could be to train the entropy model unconditionally, then freeze the representations  to train the private entropy models conditionally. The first optimization objective is not grounded in theory, and its effects should be analyzed. The goal of the Gray-Wyner Network is coding efficiency. Thus, while the purity of information is an interesting concept, we consider it to be somewhat tangential to this work.

---

> > ### Author Response · Authors · 2025-11-18
> >
> > ### The experimental tasks are relatively simple (classification, regression, segmentation). It remains unclear whether the method remains effective for tasks requiring fine-grained spatial reasoning (e.g., instance segmentation) tasks.
> >
> > In Section 4.3 (Rate-distortion performance in computer vision tasks), our experiments on the Cityscape dataset perform pixel-level semantic segmentation and depth estimation. The annotations for semantic segmentation are dense and high-quality. Although, intuitively, instance segmentation might require more spatial reasoning than semantic segmentation, we argue that the addition of the depth estimation task helps explore this aspect well enough. Depth estimation requires cues such as relative size, occlusion, linear perspective, contrast difference, and motion parallax (Burton, 1945). Many of these cues require spatial reasoning over the entire image.
> >
> > Although the COCO 2017 dataset has labels for instance segmentation, we opted to evaluate the object detection and keypoint detection pair of tasks since, intuitively, we believe they have some level of dependence but are still different enough to require some private information.
> >
> > #### References:
> >
> > Burton, Harry Edwin. The Optics of Euclid. In *Journal of the Optical Society of America*, 1945.

---

> > > ### Comment · Reviewer_czUz · 2025-11-26
> > >
> > > Thank you for the authors’ response, which has addressed most of my concerns. I have also carefully reviewed the comments from the other reviewers. From an information-theoretic perspective, disentangling representations is a direction that is worthwhile to further explore and encourage. I have raised my score, and I suggest that the authors further validate the effectiveness of their method in more practical scenarios in the future.

---

> > > > ### Author Response · Authors · 2025-12-03
> > > >
> > > > Thank you for your response and for increasing your score to 8.

---

### Official Review · Reviewer_nk8B · 2025-10-31

**Soundness:** 3
**Presentation:** 2
**Contribution:** 3
**Rating:** 4
**Confidence:** 2

**Summary:**

This paper extends the Gray–Wyner information theory framework into a learnable neural network for efficient multi-task and multi-view compression. It introduces a neural Gray–Wyner network that disentangles data into a common channel (shared information) and private channels (task-specific details), allowing controllable trade-offs between joint transmission efficiency and individual task performance. The authors derive new lossy common information bounds, connect them to Wyner’s and Gács–Körner’s classical measures, and design a trainable objective that optimizes this trade-off via a single parameter. THey experiment on synthetic data, a colorized version of MNIST,  and on real datasets including COCO and cityscapes. They show that the proposed approach achieves significant compression gains and better representation sharing compared to independent or naive joint codecs.

**Strengths:**

- Nice theoretical generalization of Grey Wyner coding to lossy compression and deep function approximators
- Nice connection between multi-task representation learning and information theory
- The authors Experiments on real world datasets like COCO and Cityscapes is appreciated and nicely complement the theoretical analysis
- Thorough theoretical analysis and high quality proofs in a detailed appendix
- Thorough reporting of architectures and parameters used

**Weaknesses:**

- A bit dense and difficult to follow, assumes alot of prereqs in info theory, perhaps consider adding an appendix of preliminaries to help readers from outside the direct field of GWC understand why this is meaningful and useful
- Though they compare against some baseline codes (joint and independant) the work exists largely outside the realm of modern computer vision representation learning and multi-task benchmarks. Some of this can be forgiven because its largely a theory paper, but some connection or comparsion to other representation learning schemes out there in the literature might make this paper more relevant for the community.
- Its unclear whether these experiments are dependant on the particulars of thenetwork architecure used, as its a bespoke set of networks and not a backbone that appears elsewhere, also not using modern tools like transformers hinders some of the relevance of the experiments.

- nit: overlapping text in Theorem 1 like 175 + 176  D_1 overlaps with Z1 hat
- Willing to improve my score

**Questions:**

It might be valuable for the community to have a better "wrap-up" of the paper. Its hard to see what this contributions enables in the space of deep learning practice that wasnt already possible with a less principled method. Does this enable predicting something fundamentally new or solving a new class of problems? Or is it just a theoretical refinement and crystalization of what practitioners have been heuristically doing.

---

> ### Author Response · Authors · 2025-11-18
> **Opening Message**
>
> We appreciate the reviewers for taking their time to analyze the paper and give thoughtful, actionable, feedback. We understand that the nature of our work can be considered niche to the ICLR community and appreciate that the reviewers took extra effort to get acquainted with it. We believe that information theory is the cornerstone of representation learning and, as such, ICLR is the best venue for our work.
>
> Please see the additional messages for our responses to each of the reviewers' comments. We apologize for the several notifications they might receive. Since some of the suggestions/weaknesses are not numbered, we decided to include them in each comment. We have submitted a PDF as supplementary material highlighting the content differences between the submitted version and the rebuttal version. The new content is highlighted in blue, whereas the erased content is highlighted in red.
>
> We hope to have addressed most concerns satisfactorily and we encourage reviewers to follow up on the discussion.

---

> ### Author Response · Authors · 2025-11-18
>
> ### A bit dense and difficult to follow, assumes alot of prereqs in info theory, perhaps consider adding an appendix of preliminaries to help readers from outside the direct field of GWC understand why this is meaningful and useful
>
> Thank you for the suggestion. Appendix A.1 (Additional preliminaries) presents the more advanced information theory concepts required to understand the theorems and their proofs. In the main paper, we deferred to a popular information theory textbook (Cover \& Thomas, 2006) for more basic concepts. As per the reviewer's suggestion, we added the mutual information and conditional mutual information concepts to Appendix A.1. To further increase self-containment of this document, please kindly point out any other concepts and we will gladly add their explanations and definitions to this section.

---

> ### Author Response · Authors · 2025-11-18
>
> ### Though they compare against some baseline codes (joint and independant) the work exists largely outside the realm of modern computer vision representation learning and multi-task benchmarks. Some of this can be forgiven because its largely a theory paper, but some connection or comparsion to other representation learning schemes out there in the literature might make this paper more relevant for the community.
>
> We are aware of several VAE-based methods in the mainstream representation learning literature that attempt to disentangle information but they are unsupervised in nature and thus unaware of tasks. Hence, we cannot fairly nor easily compare against them. We know of the well-received work of Dubois et al. (2021), but it would not be plausible to distill common information using it on our non-trivial computer vision tasks.  However, as suggested by the reviewer, we should have mentioned this work and drawn some comparisons. We added the following content to Section 2 (Previous work):
>
> > "In representation learning literature, several information-theoretic approaches suggest variational autoencoders (VAEs) that learn disentangled representations of a source (Chen et al., 2016; Higgins et al., 2017; Chen et al., 2018b). These approaches are unsupervised in nature and thus do not distill information for a specific task or isolate the common information between tasks. A more related approach (Dubois et al., 2021) proposes a channel that can achieve high performance in a set of predictive tasks, as long as they are invariant under a set of transformations. Although their proposed methods are also unsupervised, the right set of transformations can be used to isolate common information between tasks. However, finding these transformations for non-trivial tasks is an open question. Several variational methods have been proposed to measure the mutual information between two sources (Poole et al., 2019). Although they might seem useful as training objectives, these methods have significant tradeoffs between bias and variance. Moreover, they do not naturally offer the means to code (compress) the resulting representations. Compared against these existing techniques, our work directly addresses the isolation of common information between tasks and its efficient transmission."
>
> The closest work that perhaps is not well-known in the representation learning community but is so in the signal processing and compression community is the work of *coding for humans and machines* (Choi \& Bajic, 2022). However, it assumes that the tasks are nested: that the information needed for one task is embedded in the information needed for the other. Hence, only two channels are designed for two tasks: one fed to both tasks and one fed to the more complex task. The setup presented in this work generalizes that problem. The methods considered in that work, although similar in nature, would intrinsically perform poorly in ours. We covered the similarities and differences in more detail in Section 2 (Previous work).
>
> The joint and independent baseline correspond to the best possible transmit an receive rates that the experimental architecture specifics allows. Taking the architecture specifics out of the equation allows us to better evaluate the proposed architecture schema and objective function. The architecture specifics are very similar to the methods proposed in coding for humans and machines, so there is really not much motivation to compare against that work.

---

> ### Author Response · Authors · 2025-11-18
>
> ### Its unclear whether these experiments are dependant on the particulars of thenetwork architecure used, as its a bespoke set of networks and not a backbone that appears elsewhere, also not using modern tools like transformers hinders some of the relevance of the experiments.
>
> We purposefully separate the architecture schema presented in Section 3.3 (A learnable Gray-Wyner Network) from its implementation specifics presented in Section 4 (Experimental evaluation). We believe that future work can improve on the latter, with better entropy models and analysis/synthesis transforms. This is a very active research area, where inductive biases are often exploited to increase performance on specific tasks. These components form the basis of codecs, which committees such as JPEG AI and MPEG look to standardize. As such, our choices of transforms and entropy models are very well-known and established in the compression community. They have been used for image compression and several computer vision tasks (Choi \& Bajic, 2022; Foroutan et al., 2023; de Andrade \& Bajic, 2024). We are not able to use pretrained versions of them as a backbone because we need the transformations to discard information irrelevant to the tasks and, more importantly, favour the retention of common information between the two chosen tasks. Since we had to train them from scratch, the network architectures were slightly modified to better fit the requirements of our architecture scheme, our *learnable Gray-Wyner Network*. Our architecture specifics still match this previous work very closely, as pointed out in Appendix D.1 (Architecture details). Transformer-based (Zou et al., 2022; Li et al., 2024) and state-space-model (Qin et al., 2024; Zeng et al., 2025) approaches have been proposed for learnable image compression but they still have not been significantly evaluated in the context of coding for machines (compression for computer analysis tasks).
>
> The architecture schema presented in Section 3.3 (A learnable Gray-Wyner Network) has been validated by comparing it against the Separated and Combined versions, which are inspired by what has been done in the literature in the context of coding for humans and machines. In Appendix C (Theory of compatible representations), we develop some theoretical foundation and explain why our proposed schema can be better. In Section 4.1 (Transmit-receive tradeoff and ablation study), we evaluate these architecture schemas empirically. Appendix D.3 (Transmit-receive tradeoff and ablation study) shows more results, which favour the architecture schema we have chosen.
>
> Other components of the architecture schema such as the conditional entropy coding are grounded on the theory developed in Section 3.2 (Transmit-receive tradeoff optimization) and the Gray-Wyner Network, hence we do not see the need to evaluate other approaches. Some intuition as to why these choices seem to be better is presented in the last 3 paragraphs of Section 3.3 (A learnable Gray-Wyner Network).

---

> ### Author Response · Authors · 2025-11-18
>
> ### nit: overlapping text in Theorem 1 like 175 + 176 D_1 overlaps with Z1 hat
>
> Thank you for pointing this out. We fixed this overlapping text problem. We will do another pass prior to finalizing the document to make sure there are no further overlaps.

---

> ### Author Response · Authors · 2025-12-03
>
> It is unfortunate that the reviewer was not able to reply to our rebuttal due to the abrupt dissolution of the discussion period. We were hoping for an increase in their score since the reviewer mentioned their potential willingness to do so.

---

### Meta-Review · Area_Chair_zU5v · 2025-12-22

**Summary:**

This paper generalizes the Gray–Wyner information-theoretic framework to a learnable neural architecture for multi-task and multi-view compression. It introduces a neural Gray–Wyner network that separates data into a shared common representation and task-specific private representations, enabling a controllable balance between overall compression efficiency and individual task accuracy. The authors derive new bounds on lossy common information, relate them to the classical notions of Wyner and Gács–Körner common information, and propose a unified training objective with a single tuning parameter to manage this trade-off. Experimental results on synthetic data, colorized MNIST, and real-world datasets such as COCO and Cityscapes show that the method achieves superior compression efficiency and more effective representation sharing than independent or naive joint coding baselines.

**Reviewer Concerns:**

The following concerns are still outstanding

(1) Difficult to follow

(2) Baselines are outdated

(3) Lack of error bar.

**Reviewer Scores:**

Reviewer nk8B (score: 4, low confidence) indicated a willingness to increase the score and is likely to raise it to 6.

Reviewer czUz (score: 6) increased the score to 8.

Reviewer wYFu (score: 6) indicated an intention to maintain the original score.

---

### Decision · Program_Chairs · 2026-01-26

Accept (Poster)